# The hidden side of animal cognition research: Scientists' attitudes toward bias, replicability and scientific practice

Benjamin G. Farrar[1,2]*, Ljerka Ostojić[3], Nicola S. Clayton[1]

1 Department of Psychology, University of Cambridge, Cambridge, United Kingdom, 2 Institute for Globally Distributed Open Research and Education, United Kingdom, 3 Faculty of Humanities and Social Sciences of Rijeka, University of Rijeka, Rijeka, Croatia

* bgf22@cam.ac.uk, farrarbg@gmail.com

**Data Availability Statement:** All survey response data are openly available at https://osf.io/6j7kp/. To protect the identity of the researchers who completed the survey, the demographic data is not openly available.

## Abstract

Animal cognition research aims to understand animal minds by using a diverse range of methods across an equally diverse range of species. Throughout its history, the field has sought to mitigate various biases that occur when studying animal minds, from experimenter effects to anthropomorphism. Recently, there has also been a focus on how common scientific practices might affect the reliability and validity of published research. Usually, these issues are discussed in the literature by a small group of scholars with a specific interest in the topics. This study aimed to survey a wider range of animal cognition researchers to ask about their attitudes towards classic and contemporary issues facing the field. Two-hundred and ten active animal cognition researchers completed our survey, and provided answers on questions relating to bias, replicability, statistics, publication, and belief in animal cognition. Collectively, researchers were wary of bias in the research field, but less so in their own work. Over 70% of researchers endorsed Morgan's canon as a useful principle but many caveated this in their free-text responses. Researchers self-reported that most of their studies had been published, however they often reported that studies went unpublished because they had negative or inconclusive results, or results that questioned "preferred" theories. Researchers rarely reported having performed questionable research practices themselves —however they thought that other researchers sometimes (52.7% of responses) or often (27.9% of responses) perform them. Researchers near unanimously agreed that replication studies are important but too infrequently performed in animal cognition research, 73.0% of respondents suggested areas of animal cognition research could experience a 'replication crisis' if replication studies were performed. Consistently, participants' free-text responses provided a nuanced picture of the challenges animal cognition research faces, which are available as part of an open dataset. However, many researchers appeared concerned with how to interpret negative results, publication bias, theoretical bias and reliability in areas of animal cognition research. Collectively, these data provide a candid overview of barriers to progress in animal cognition and can inform debates on how individual researchers, as well as organizations and journals, can facilitate robust scientific research in animal cognition.

**Funding:** BGF was supported by the University of Cambridge BBSRC Doctoral Training Programme (BB/M011194/1).

**Competing interests:** The authors have declared that no competing interests exist.

## Introduction

Animal cognition research covers a wide range of topics, from how animals learn and remember to how they make decisions and how they interact with other individuals. By studying a wide number of questions in an equally wide range of species, the field broadly aims to understand the mechanisms, functions and the evolution of cognition (although exact definitions depended on context, for example see [1–3] for 'comparative cognition' and for 'comparative psychology' see [4,5]). However, studying animal minds—that are in principle unobservable—is challenging [6,7], and the process is shaped by a variety of assumptions about minds, animals, and knowledge [8], as well as the history of the field itself. Notably, throughout this history, animal cognition researchers have often sought to improve how we study animal cognition. They foregrounded debates on experimenter bias [9,10], parsimony [11], and ecological validity [12], and the importance of a wide range of anecdotal [13,14], observational [15] and experimental data [16,17]. However, whether the field has vastly improved its methods along these lines is debated. Concerns about experimenter bias [18,19], parsimony [20,21], and validity [22–25] still dominate the literature today, and concerns have also been raised about the reliability of the statistical effects that are reported in the animal cognition literature [26–28].

The extent to which some areas of animal cognition are making progress is hotly debated [29–39]. However, these debates are often performed by a minority of stakeholders in animal cognition research—often between those who claim discoveries of "higher" processes in animals and their corresponding 'killjoys' or skeptics, accompanied by a meta-commentary from a small number of interested researchers and philosophers. But how effectively these debates are reaching animal cognition researchers in general, and how they are received, has garnered little attention. Survey studies can address this by directly asking researchers their opinions on key debates in the field, how their own research practices are shaped by these debates, and what they feel is incentivised in academia. For example, survey studies have quantified the negative effects on researchers' mental health due to academia's "publish or perish" culture [40], and researchers often report that scientific incentives are misaligned with their scientific ideals. For example, ecology researchers reported that while they thought replication studies were a crucial use of resources, they experienced difficulty obtaining funding for them and, even if they were performed, they perceived barriers to publishing them [41]. More directly, researchers have self-reported using false-positive inflating research practices at non-negligible rates [42–44], and also measured editor and reviewer biases against replication studies [45,46].

In the current study, we surveyed researchers' attitudes towards several contentious topics in animal cognition research and examined how their own and others' research methods might be affected by these debates and wider scientific incentives. Thus, the survey was designed to, i) survey the extent to which researchers are concerned about certain research and publication practices in the field, ii) collect direct evidence of the rates of these practices from researchers themselves, practices that may otherwise be difficult to observe, and iii) provide researchers with the opportunity to voice any concerns or opinions they have about how animal cognition research operates. These data may impact the field in three ways. Firstly, they can help researchers critically evaluate the evidential strength of published findings, given how frequently researchers estimate certain biases to be present. Secondly, they can facilitate debates on the effectiveness of the scientific process in animal cognition and engage researchers and students in these debates. Finally, they can help to identify barriers to effective scientific research of animal cognition that can inform policy making in journals, funding bodies and hiring committees, as well as decision making by individual animal cognition researchers.

We invited 1001 researchers who have published in animal cognition journals in the last three years to answer a range of questions about bias and research practices in animal cognition research. The survey consisted of five blocks of questions, broadly covering, i) bias, ii) publication practices, iii) statistics, iv) replication, and v) how researchers derive their own beliefs about animal cognition. These five blocks were not mutually exclusive (e.g., answers about "bias" featured throughout), but were loosely based on some of the key challenges facing animal cognition research, and science more broadly [1,7,47–49].

The full survey is presented in the Methods section, but briefly, the five blocks covered topics as follows. The **Bias** block asked researchers about experimenter bias and objectivity in their own work, and about the role bias might play in shaping the results and theories in animal cognition research more broadly. The final topic of the bias block was Morgan's canon—the notion that animal behaviour should not be interpreted in terms of "higher" psychological processes if it can be fairly interpreted in terms of "lower" processes—with researchers answering whether they agreed that "Morgan's canon is important to use when interpreting the results of animal cognition research". The **Publication** block first asked to what extent researchers thought that they themselves, and other researchers, make appropriate claims when submitting research for publication. Second, as a direct measure of publication bias, we asked researchers which proportion of their own studies has been published, or will be published for ongoing studies, as well as the reasons why some of their studies go unpublished. The **Statistics** block then measured researchers' confidence in their own statistical analyses, and their ability to judge the validity of other analyses. Next, it asked researchers to estimate the prevalence of "questionable research practices", which may increase the likelihood of spurious findings in their own and in others' research. The **Replication** block first focused on attitudes towards replication studies; how important are replications, and are replications performed often enough in their own area of research and others? Second, it asked researchers about whether they believe their own area of research, or other areas of research in animal cognition, would experience a 'replication crisis' if multiple replication studies were attempted, and how many of these replication studies they would predict to be 'successful'. Finally, the **Belief** block asked researchers a range of questions about how they decide what to believe about animals' cognition. We asked researchers about the role that scientific experiments and day-to-day experience play in shaping these beliefs, as well as how often they agree with the conclusions presented in scientific papers.

## Materials and methods

We invited all researchers who are a first, last or corresponding author on any type of article published in the past three years (i.e., 2018–2020 inclusive) from the following six animal cognition journals to complete our survey: *Animal Cognition*, *Animal Behavior and Cognition*, *Journal of Comparative Psychology*, *International Journal of Comparative Psychology*, *Journal of Experimental Psychology*: *Animal Learning and Cognition*, *Frontiers in Psychology*: *Comparative Psychology*. BGF viewed every article from these journals between 2018 and 2020, and extracted the email addresses of the first, last and corresponding authors. If these email addresses were not provided in the article, BGF conducted a keyword-based web search to try to find one for the author in question. In total, 1161 authors were identified and email addresses for 1004 of these could be located from the articles or web searches. Of these, three email addresses were our own, leaving a final sample of 1001. Emails were sent to these 1001 researchers in January 2021. Sixty-four emails returned errors, and BGF conducted further web searchers to identity alternative emails for these researchers, of which 32 were obtained

and the survey invite emailed to. Of the 969 successfully sent emails, 210 completed surveys were returned (response rate = 21.6%).

Researchers completed a questionnaire hosted on Qualtrics. The study protocol was approved by the University of Cambridge's Psychology Research Ethics Committee (PRE.2020.096). The survey was designed by BGF, with feedback from NSC and LO, and then piloted on several volunteers from the Comparative Cognition Laboratory at the University of Cambridge. The full survey is detailed below, and the anonymized survey data and analysis code are available at osf.io/6j7kp.

## Demographics

Participants gave informed consent and answered the following demographic questions:

1. On approximately how many papers have you been an author or co-author about animal learning, cognition, behavior or welfare?

2. Approximately how many years have you worked in animal learning, cognition, behavior or welfare?

3. How well do each of these terms describe your research? (Not at all, Slightly, Moderately, Very, Extremely)

   *Animal behavior, animal cognition, animal learning, behavioral ecology, behavioural neuroscience, comparative psychology*

4. In which of these journals have you authored or co-authored a paper?

   *Animal Cognition, Animal Behavior and Cognition, Journal of Comparative Psychology, International Journal of Comparative Psychology, Journal of Experimental Psychology: Animal Learning and Cognition, Frontiers in Psychology: Comparative Psychology.*

Researchers then completed five blocks of questions about research methods and their attitudes to various issues facing animal cognition research. The order of the blocks was randomized between participants. Each block was presented on one page of the survey, with a line separating between the first set and second set of questions in each block. The exact questions and formatting of each block are presented in Figs 1–5.

## Bias

This block contained 7 questions about bias in animal cognition research. Questions 1 to 3 asked about researchers' attitude towards bias in their own research: whether they hoped for particular results when performing experiments; whether they are concerned that they might bias the results of their studies towards certain results and whether they thought they could detach from any biases to perform objectively fair tests of animal cognition.

Questions 4–7 asked about bias across animal cognition research: whether they think the results and theories in their own area, and in other areas, of animal cognition are strongly affected by researchers' biases, and whether, if they knew the topic and the authors, they would be able to guess the conclusions of a published study without reading it. Question 7 asked whether they thought that Morgan's canon is important to use in animal cognition research, and, before answering the question, participants read the following introductory text: Morgan's canon states that: *"In no case is an animal activity to be interpreted in terms of higher*

This section involves questions about **publications** in animal cognition.

When publishing a paper, researchers make claims about what their data mean.

When submitting a paper for peer review, I tend to:

○ make weaker claims than are warranted by the data
○ make appropriate claims given my data
○ make stronger claims than are warranted by the data
○ N/A

After peer review, my claims tend to:

○ become weaker
○ stay the same
○ become stronger
○ N/A

When submitting papers, I believe that **other researchers** tend to:

○ make weaker claims than are warranted by the data
○ make appropriate claims given their data
○ make stronger claims than are warranted by the data

Publication bias occurs when the decision to publish studies is dependent on the results of the studies, e.g., when negative results are not published.

What **percent** of the studies that you have performed have been published and/or you think will be published?

| 0 | 10 | 20 | 30 | 40 | 50 | 60 | 70 | 80 | 90 | 100 |

% Of studies

In the past, have you been unable to publish a study you have performed, or have you ever decided not to publish a study?

○ Yes
○ No
○ N/A

Why did you choose not to publish the study, or why do you think the study was not published?

Do you have any other comments about **publication** in animal cognition?

**Fig 1. The bias questions.**

This section asks about **bias** in animal cognition.

|  | Never | Rarely | Sometimes | Often | Always | N/A |
|---|---|---|---|---|---|---|
| When performing research, I find myself hoping for one result over others. | ○ | ○ | ○ | ○ | ○ | ○ |
| I am concerned that I might bias the results of my studies towards certain results. | ○ | ○ | ○ | ○ | ○ | ○ |
| I can detach from any biases to perform objectively fair tests of animal cognition | ○ | ○ | ○ | ○ | ○ | ○ |

|  | Strongly Disagree | Disagree | Neither agree nor disagree | Agree | Strongly Agree | N/A |
|---|---|---|---|---|---|---|
| The results and theories **in my area** of animal cognition are strongly affected by researchers' biases | ○ | ○ | ○ | ○ | ○ | ○ |
| The results and theories **in other areas** of animal cognition are strongly affected by researchers' biases | ○ | ○ | ○ | ○ | ○ | ○ |
| If I knew the topic and the authors, I would be able to guess the conclusions of a published study without reading it | ○ | ○ | ○ | ○ | ○ | ○ |

Morgan's canon states that: "In no case is an animal activity to be interpreted in terms of higher psychological processes if it can be fairly interpreted in terms of processes which stand lower in the scale of psychological evolution and development."

|  | Strongly disagree | Somewhat disagree | Neither agree nor disagree | Somewhat agree | Strongly agree |
|---|---|---|---|---|---|
| Morgan's canon is important to use when interpreting the results of animal cognition research | ○ | ○ | ○ | ○ | ○ |

Do you have any other comments about **bias** in animal cognition research?

**Fig 2. The publication questions.**

This section involves questions about **replications** in animal cognition research.

Consider 100 typical papers in your research area. What percentage of studies would successfully replicate the original results, if the same protocol was used on a new sample of similar size in the same species (assuming that this was possible)?

Use whichever definition of successful you think is appropriate.

| 0 | 10 | 20 | 30 | 40 | 50 | 60 | 70 | 80 | 90 | 100 |
|---|----|----|----|----|----|----|----|----|----|-----|

% Successful

Please indicate the percentage of studies you think would successfully replicate the original results if replication studies had a **sample size of 1000 animals**?

| 0 | 10 | 20 | 30 | 40 | 50 | 60 | 70 | 80 | 90 | 100 |
|---|----|----|----|----|----|----|----|----|----|-----|

% Successful

|  | Strongly disagree | Somewhat disagree | Neither agree nor disagree | Somewhat agree | Strongly agree | N/A |
|--|-------------------|-------------------|----------------------------|----------------|----------------|-----|
| **My area** of animal cognition research would experience a "replication crisis" if attempts to replicate most of its studies were conducted. | ○ | ○ | ○ | ○ | ○ | ○ |
| **Some areas** of animal cognition research would experience a "replication crisis" if attempts to replicate most of its studies were conducted. | ○ | ○ | ○ | ○ | ○ | ○ |
| I could identify animal cognition studies that would successfully replicate and those which would not. | ○ | ○ | ○ | ○ | ○ | ○ |

|  | Strongly disagree | Disagree | Neither agree nor disagree | Agree | Strongly agree | N/A |
|--|-------------------|----------|----------------------------|-------|----------------|-----|
| It is important that replication studies are performed in animal cognition research. | ○ | ○ | ○ | ○ | ○ | ○ |
| Enough replication studies are published **in my area** of animal cognition. | ○ | ○ | ○ | ○ | ○ | ○ |
| Enough replication studies are performed in animal cognition in general. | ○ | ○ | ○ | ○ | ○ | ○ |

Do you have any other comments about **replication** in animal cognition research?

**Fig 3. The replicability questions.**

This section involves questions about **statistics** in animal cognition research.

| | Strongly disagree | Somewhat disagree | Neither agree nor disagree | Somewhat agree | Strongly agree | N/A |
|---|---|---|---|---|---|---|
| When I perform a statistical analysis, I know that the analysis is valid and appropriate. | O | O | O | O | O | O |
| I could explain why my analysis is appropriate and valid to another researcher. | O | O | O | O | O | O |

| | Never | Rarely | Sometimes | Often | Always |
|---|---|---|---|---|---|
| When I read or review an article in my area of animal cognition, I can assess whether the statistical methods used are valid and appropriate. | O | O | O | O | O |

There is growing concern that researchers use false positive inflating research practices in science, such as:

- Performing many analyses and selectively reporting the statistically significant ones
- Reporting an unexpected finding as if it was predicted from the start
- Data dredging/p-hacking/fishing for significance
- Selectively excluding data points to produce a significant/desired result
- Collecting more data until a significant/desired result is obtained

How common do you think these research practices are in...

| | Never | Rare | Sometimes | Often | Always | N/A |
|---|---|---|---|---|---|---|
| Your own research | O | O | O | O | O | O |
| Other research in animal cognition | O | O | O | O | O | O |

Do you have any other comments about **statistics** in animal cognition?

Fig 4. The statistics questions.

This section involves questions about your **beliefs** about the cognition of animals.

|  | Strongly disagree | Somewhat disagree | Neither agree nor disagree | Somewhat agree | Strongly agree |
|---|---|---|---|---|---|
| The results of scientific experiments affect my beliefs about the cognition of animals. | ○ | ○ | ○ | ○ | ○ |
| My day-to-day experience interacting with animals affects my beliefs about the cognition of animals. | ○ | ○ | ○ | ○ | ○ |

Please indicate the extent to which your beliefs about the cognition of animals are determined by the scientific literature or by your day-to-day experience interacting with animals:

Exclusively by science                                    Exclusively by experience

| 0 | 10 | 20 | 30 | 40 | 50 | 60 | 70 | 80 | 90 | 100 |

|  | Never | Rarely | Sometimes | Often | Always | N/A |
|---|---|---|---|---|---|---|
| When I read a paper in **my area** of animal cognition research, I agree with the authors' interpretation of their data | ○ | ○ | ○ | ○ | ○ | ○ |
| When I read a paper in **other areas** of animal cognition research, I agree with the authors' interpretation of their data | ○ | ○ | ○ | ○ | ○ | ○ |

Do you have any other comments about your beliefs about the cognition of animals and the role of science in forming them?

**Fig 5. The belief questions.**

*psychological processes if it can be fairly interpreted in terms of processes which stand lower in the scale of psychological evolution and development."*

Question 8 then asked researchers for any other comments about the Bias questions.

## Publication

This block contained 7 questions on publication practices in animal cognition research. All questions had an NA option, which we have excluded here for brevity. The first 3 questions focused on the claims that researchers make when publishing papers, and the last 4 questions focused on publication bias, asking researchers the percent of their studies that are, or will be, published, and the reasons some of their studies go unpublished. Question 7 asked researchers for any other comments about the Publication questions.

## Replicability

This block contained 9 questions on researchers' attitudes towards replicability in animal cognition research. Questions 1 and 2 concerned the likely success of replication studies in the researcher's area of animal cognition research, and Questions 3 and 4 asked researchers whether their own, or some, areas of animal cognition research would experience a "replication crisis" if attempts to replicate most of its studies were conducted. Question 5 then asked whether researchers thought they could identify animal cognition studies that would successfully replicate and those which would not. Questions 6 to 8 next asked about the importance and frequency of replications in the field, Question 9 then asked researchers for any other comments about the Replication questions.

## Statistics

This block contained 6 questions on the use of statistics in animal cognition research. Questions 1 and 2 concerned the confidence researchers have in their understanding of their own statistical analyses, Question 3 focused on their self-reported understanding of others' statistical analyses, and Questions 4 and 5 then asked about the prevalence of questionable research practices [44] in animal cognition research. Question 6 then asked researchers for any other comments about the Statistics questions. Before Questions 4 and 5, participants read a brief description of some questionable research practices:

> There is growing concern that researchers use false positive inflating research practices in science, such as:
>
> Performing many analyses and selectively reporting the statistically significant ones
>
> Reporting an unexpected finding as if it was predicted from the start
>
> Data dredging/p-hacking/fishing for significance
>
> Selectively excluding data points to produce a significant/desired result
>
> Collecting more data until a significant/desired result is obtained

## Belief

This block contained 6 questions on how researchers derive their beliefs about animal cognition. Questions 1, 2 and 3 asked about the role of scientific research and researchers' day-to-day experience in forming their beliefs about animal cognition, and Questions 4 and 5 asked how often researchers agree with the claims made in scientific papers. Finally, Question 6 asked researchers for any other comments about the Belief questions.

## Free-text analysis

Throughout the results, we provide direct quotes of participants' answers to the free-text responses. These quotes were taken from participants who, at the end of the survey, opted in for their free-text answers to be shared openly and were screened for any identifying information. If a free-text response contained clearly identifying information, it was excluded from the open data-set. All the free-text answers for which we received consent to share, and which did not contain identifying information are openly available at osf.io/6j7kp. In addition to directly quoting partici-pants' free-text answers, of which only a minority could be included in the report, we also catego-rized their free-text responses based on the common themes that they included within each block. First, one author (BGF) read through all responses and identified common themes in participants' responses. He then marked whether each response fit each category or not. If a response matched more than one category, this was still recorded, i.e., a single response could in principle fit all the categories. A second author (LO) was given the category descriptions and, blind to the first coder's decisions, also marked whether each response fit each category or not. Of BGF's 481 decisions to label a response with a category, LO independently agreed with 402 (83.6%) of them. In addition, LO made 103 classifications that BGF had not originally and suggested four further category labels, three of which were retained. Each disagreement was resolved by discussion between BGF and LO, with the most disagreements either being an error from one of the two coders originally, or cases where both coders agreed that the statement was ambiguous, i.e., there were no cases of disagreement that could not be resolved through discussion. Our category-based analyses are pre-sented for the Publication, Statistics, Replication and Belief blocks. For the Bias block, we chose to split the results of the open-ended question ("Do you have any other comments about bias in ani-mal cognition research?") into two tables, as participants' free-text responses were split between providing examples of biases in animal cognition research and elaborating on their Likert-type responses to the question about Morgan's canon. In addition to our category based-analysis, we also present some quotes in-text that we felt highlighted an important topic that our category-based analysis might have missed. Where some themes occurred across blocks of topics but were not necessarily directly related to the topic in question, we present these in a "miscellaneous" sec-tion, although this was not performed systematically.

## Results

### Demographics

From 1001 invitations, we received 210 completed surveys (response rate = 21.6%). Our sample of researchers had published a median of 17 papers on topics in animal cognition (IQR: 8–50) and had been active in the field for a median of 14 years (IQR: 8–25). Table 1 displays these

**Table 1. The number of papers published and years active in animal cognition of the 210 researchers completing the survey.**

| Number of papers | 0 | 1–5 | 6–10 | 11–25 | 25–75 | > 75 |
|---|---|---|---|---|---|---|
| % | 0.4 | 17.6 | 22.8 | 19.5 | 24.8 | 14.8 |
| N | 1 | 37 | 48 | 41 | 52 | 31 |
| **Years active** | **0** | **1–3** | **3–7** | **8–15** | **15–25** | **>25** |
| % | 0.4 | 2.3 | 18.1 | 28.1 | 21.4 | 20.5 |
| N | 1 | 5 | 38 | 59 | 45 | 43 |

One response for years active was left blank and therefore excluded. The one researcher who reported 0 for papers published and years active later described publishing in at least one of our target articles, suggesting that the 0 responses may have been in error.

demographics. An exploratory k-means cluster analysis of participants' endorsement of key words describing their research suggested that we had two main groups of participants completing the survey—a larger group of researchers in animal cognition and comparative psychology, and a smaller group of behavioural ecologists (see Supporting Information for details).

## Bias

We asked researchers about bias in their own experiments, and their perceptions of bias across the field. Researchers frequently reported either sometimes (39.7% of respondents) or often (38.8%) hoping for one result over another when performing research, and researchers were split between either being rarely concerned (36.5%) or sometimes concerned (30.3%) that they might bias the results of their studies towards a certain conclusion. Nevertheless, they reported that they could often (45.8%) or always (38.4%) detach from any biases to perform objectively fair tests of animal cognition (Fig 6).

In terms of bias across the field, researchers were split between agreeing (29.6%), disagreeing (23.8%) and neither agreeing nor disagreeing (36.4%) that the results and theories in their own area of animal cognition are strongly affected by researchers' biases. Responses were similar when researchers were asked to consider bias in other areas of animal cognition, but more researchers agreed that the results and theories are strongly affected by researchers' biases (agree: 36.0%; neither agree nor disagree: 39.0%; disagree 14.5%). Researchers were split between agreeing (34.0%), disagreeing (22.3%) or neither agreeing or disagreeing (30.6%) that if they knew the topic and the authors, they would be able to guess the conclusions of a study without reading it (Fig 7). Notably, most respondents tended to avoid the extreme responses—no more than 10.5% of respondents chose the strongly agree or strongly disagree for these questions on bias.

We received 68 free-text responses concerning bias in the field, many of which elaborated on the question about Morgan's canon. However, researchers reported a diverse range of attitudes towards bias in the field. While most researchers reported they could detach from their own biases readily on the Likert-measure (Fig 6), perhaps through using measures such as blinding, other researchers expressed skepticism about the ability to perform research objectively:

> "As to the first three questions on my own bias—it is NEVER possible to detach yourself from your own biases. You can only try your best and take as many steps as possible to control for this, which I do. . . As to hoping for one result over another—as negative results are unpublishable, any sane scientist will hope for positive results. Our careers, and often our livelihoods, rely on getting positive results and publishing them. Too much is at stake to pretend that there is no bias."

Researchers indicated several different forms of bias that might affect animal cognition research, ranging from anthropomorphism and confirming "higher" abilities in animals, to excessive skepticism. Table 2 presents a selection of these reported biases.

## Morgan's canon, simplicity, and parsimony

We next asked researchers about the role of Morgan's canon. Most researchers agreed somewhat (38.6%) or strongly (31.9%) that Morgan's canon is important to consider when interpreting the results of animal cognition research (Fig 8). However, researchers often elaborated on these answers in the free text responses, revealing a more nuanced perspective of the use of Morgan's canon, which are detailed in Table 3.

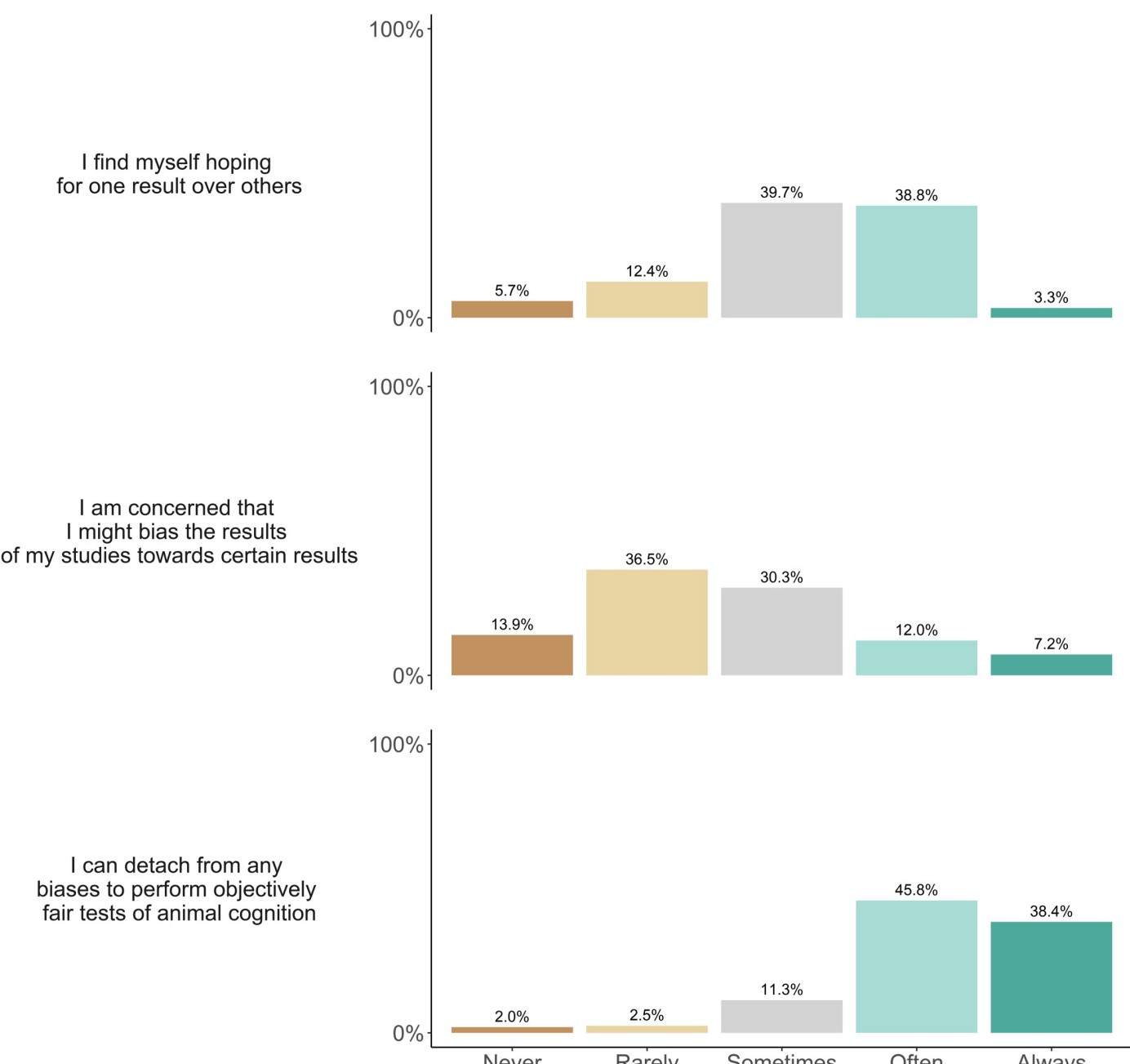

**Fig 6. Animal cognition researchers' self-reported concern about bias in their own studies (N = 210).** Percentages may not add to 100% due to a small number of NA responses.

## Publication

We asked researchers whether they believe themselves and others to make appropriate claims when submitting research for publication, and how many of their studies end up being published. When submitting papers for publication, 86.0% of researchers reported that they make

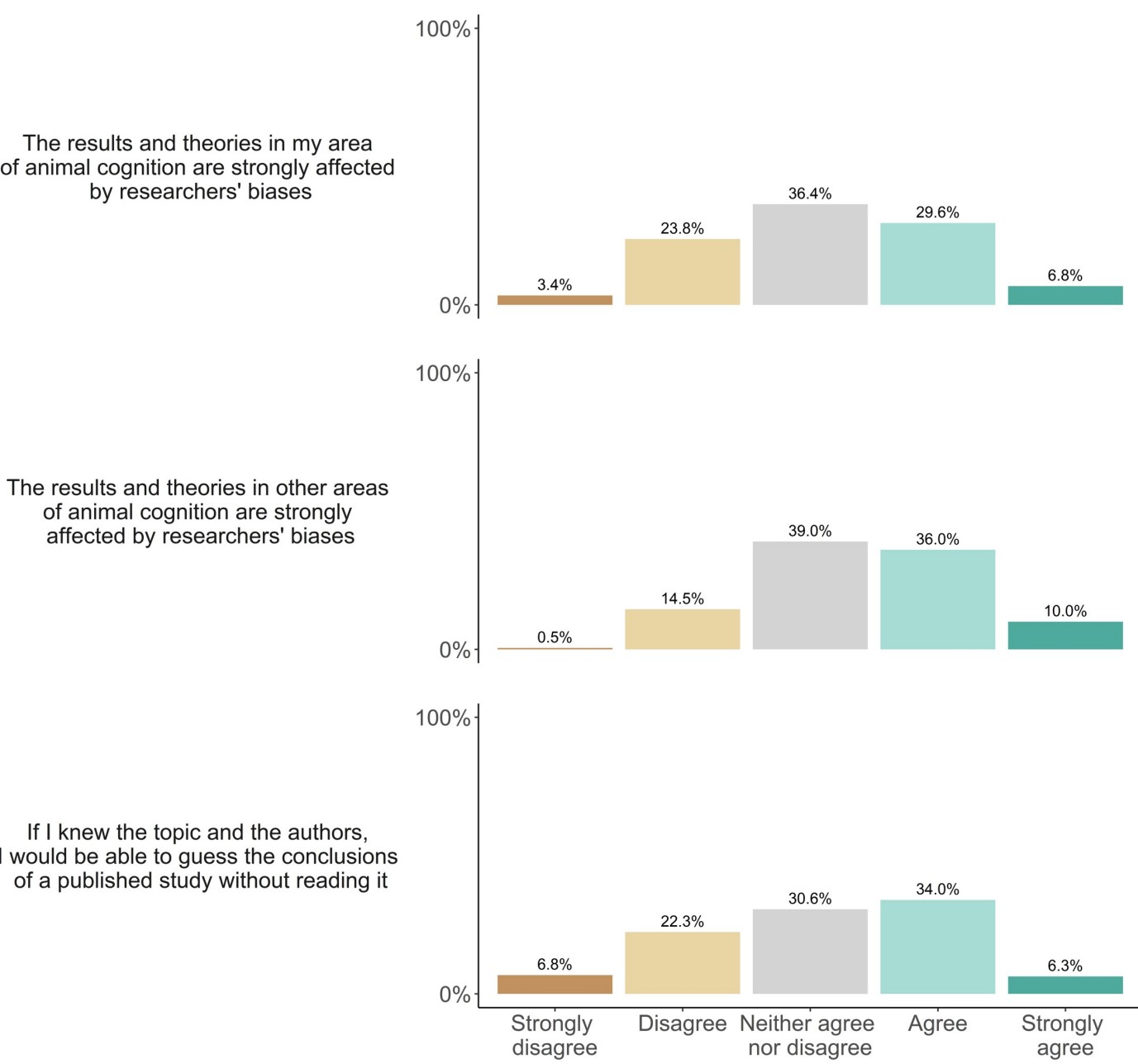

**Fig 7. Animal cognition researchers' self-reported concern about bias in animal cognition research (*N* = 210).**

appropriate claims given their data, while only a small number stated that they overclaim (7.7%) or underclaim (5.8%). In contrast, our sample was split between believing that other researchers were likely to make stronger claims than warranted by their data (56%), and believing that others make appropriate claims (43%, Fig 9). Researchers reported that their own claims usually stayed the same (69.0%) or became weaker (21.0%) after peer review. A minority of researchers reported that their claims increased in strength (9.0%). When asked how

Table 2. Animal cognition researchers' beliefs about bias in animal cognition research.

| Do you have any other comments about bias in animal cognition research? | N | Exemplars |
|---|---|---|
| Provided an example of bias | 31 | *"Bias is of two kinds: (a) bias against animals in comparison with humans (pro-human bias) and (b) bias to interpret animal behaviour as evidence for complex cognition (pro-animal bias). Both kinds of bias undermine the legitimacy of animal cognition research."* |
| Suggested that bias is predominantly in how the data are interpreted | 30 | *"The bias I typically see is a bias to produce a narrative that goes beyond the data, I am not sure whether that bias goes into the design/approach that produces the data. I almost never have concerns that design/data might be unethically tweaked (i.e., conscious bias)."* |
| Suggested that bias is inherent to many study designs | 29 | *"It is often in the selection of the behavioral markers that the biases are most strongly evident, so the biases are well-entrenched well before data analysis. Given the modern trend to only provide heuristic descriptions of what was measured and reliance on inter-observer reliability to justify those measurements means that the rationale used for the selection process is usually hidden from the reader. This makes it difficult to identify the implicit biases in the study."* |

many of their studies had, or for ongoing studies, will, end in publication, the median response was 80% (IQR: 70% - 90%, Fig 10). However, there was a large spread in responses, with 23 respondents saying 50% or fewer of their studies have been published, and 17 reporting that they have published all of their studies.

We received 144 free-text responses from researchers explaining why certain studies of theirs had not been published. The responses suggested several different causes of publication bias in the field. Some researchers reported self-filtering studies they deemed of little importance:

"*I have a few studies that are just not adequate to publish, in terms of experimental design, subject size, or no informative findings (and I'm including null results as potentially informative). These are my own issues, not that of the publication process.*"

Another reported cause of self-driven publication bias was a lack of incentives to publish all research, either due to time constraints or perceptions of how publishing all work would affect funding opportunities:

"*My position is dependent on grant funding, this contingency is coercive to publishing only the studies that strengthen the grant.*"

Although not one of our identified themes, sixteen researchers (11% of free-text responses) also reported that publication bias was enforced by journals, reviewers and editors:

"*Consistent rejection across journals, which typically reported that the findings were not "attractive enough" (e.g., replications, inconclusive results, etc.)*"

Through our categorization analysis, the most common themes we identified were articles not being published for containing inconclusive results (31), design limitations (30), negative results (29), insufficient resources for publication (29) and too few data (28). In Table 4, we highlight quotes from each of the 10 themes we identified in the responses. Next, Table 5

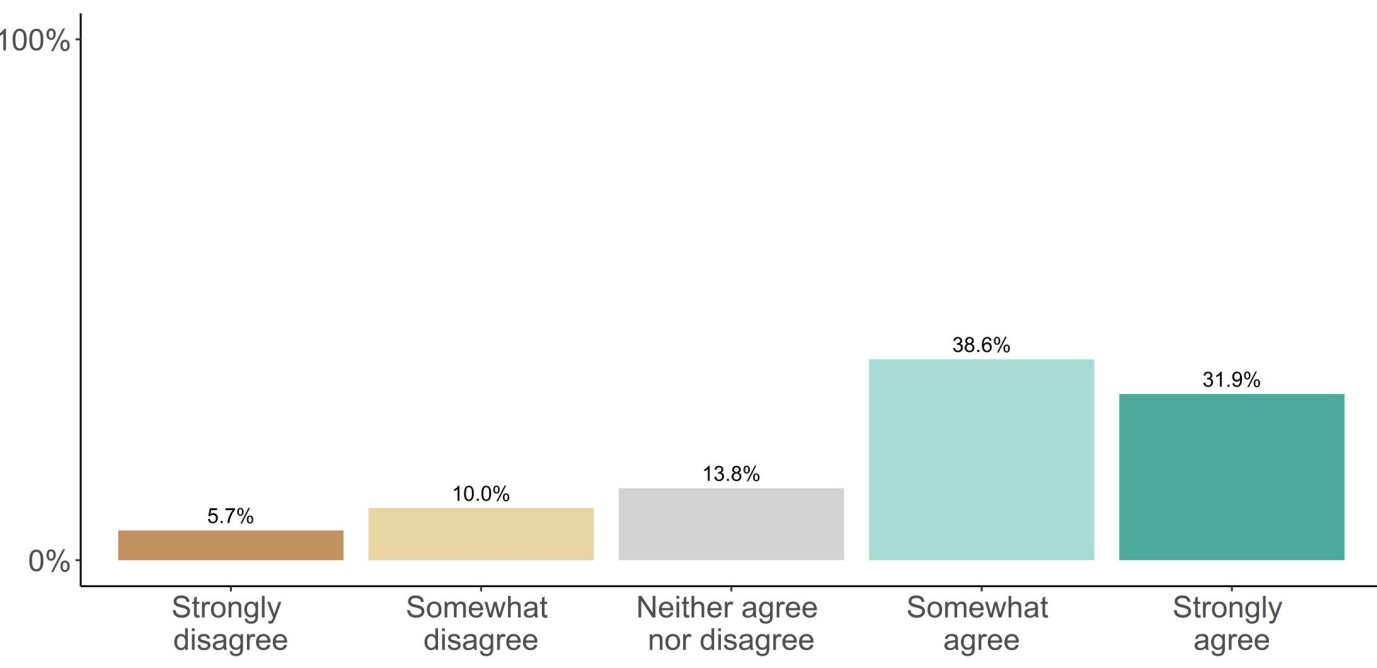

**Fig 8. Animal cognition researchers' endorsement of Morgan's canon (N = 210).**

**Table 3. Animal cognition researchers' attitudes towards Morgan's canon as a tool in animal cognition research.**

| Do you have any other comments about bias in animal cognition research? Answers that referenced Morgan's canon | N | Exemplars |
|---|---|---|
| Caveated or criticized the use of Morgan's Canon | 16 | *"In my opinion, Morgan's canon often leads to more bias rather than less. As scientists, it is important to have an open mind in both directions. E.g. looking at the evolutionary tree of a species when interpreting its behaviour is often more conclusive that Morgan's canon."* <br> *"I think that biases on the cognitive processes underlying certain behaviours can go both ways. One could overstate the complexity, as much as one could underestimate it. That is why in general I do not consider Morgan's canon to be always useful: to use it best, we would need to have a clear understanding of what process "stand lower in the scale of psychological evolution" without pre-existing biases."* |
| Suggested that parsimony is important when interpreting data | 30 | *"In terms of Morgan's canon, it is not dissimilar to parsimony in phylogenetics or Occam's razor in normal scientific inquiry. Showing skepticism in cause does not suggest that more complicated cognitive explanations exist, but the onus is on the researcher to demonstrate."* <br> *"While Morgan's canon is useful as a philosophical tool, I do think that it often conflicts with parsimony arguments made from phylogeny, so in practice I feel it often does not help per se."* |

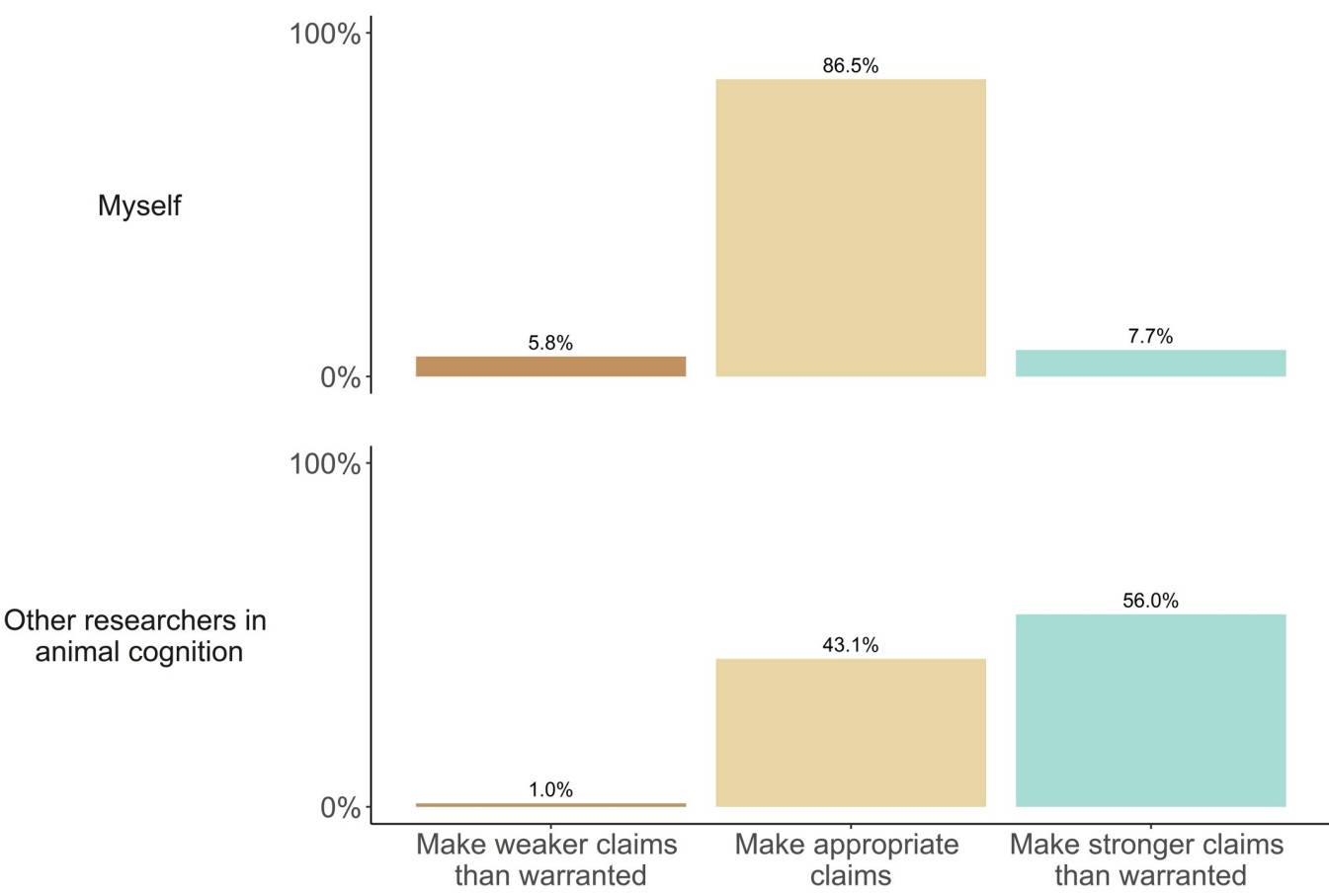

**Fig 9. Animal cognition researchers' beliefs about overclaiming and underclaiming when submitting research articles for publication, N = 210.**

highlights several quotes from the open-ended free-text question about publication practices in animal cognition.

## Statistics

We asked researchers about their confidence in their own statistical analyses, their ability to assess others' analyses, and the rate of questionable research practices in the field. Researchers strongly or somewhat agreed that when they perform a statistical analysis, they know it is appropriate and valid (strongly agree: 53.2%, somewhat agree 42.9%), and that they could explain why this was the case to another researcher (strongly agree: 59.5%, somewhat agree 36.6%, Fig 11). When reading or reviewing others' research, our sample reported that they could often (59.8%), sometimes (23.4%) or always (12.4%) assess the validity of the analysis. A minority of researchers reported that they could rarely (3.8%), or never (0.5%) assess the validity of the analysis. When asked how often they themselves or other researchers performed questionable research practices (QRPs), which may induce false positive findings, researchers reported that they themselves rarely (41.1%), never (31.2%), or sometimes (20.3%) conducted QRPs. However, researchers thought that others either sometimes (52.7%), often (27.9%), or rarely (18.4%) did so (Fig 12). We received 66 free-text responses about the use of statistics in

## What proportion of your studies have been, or you think will be, published?

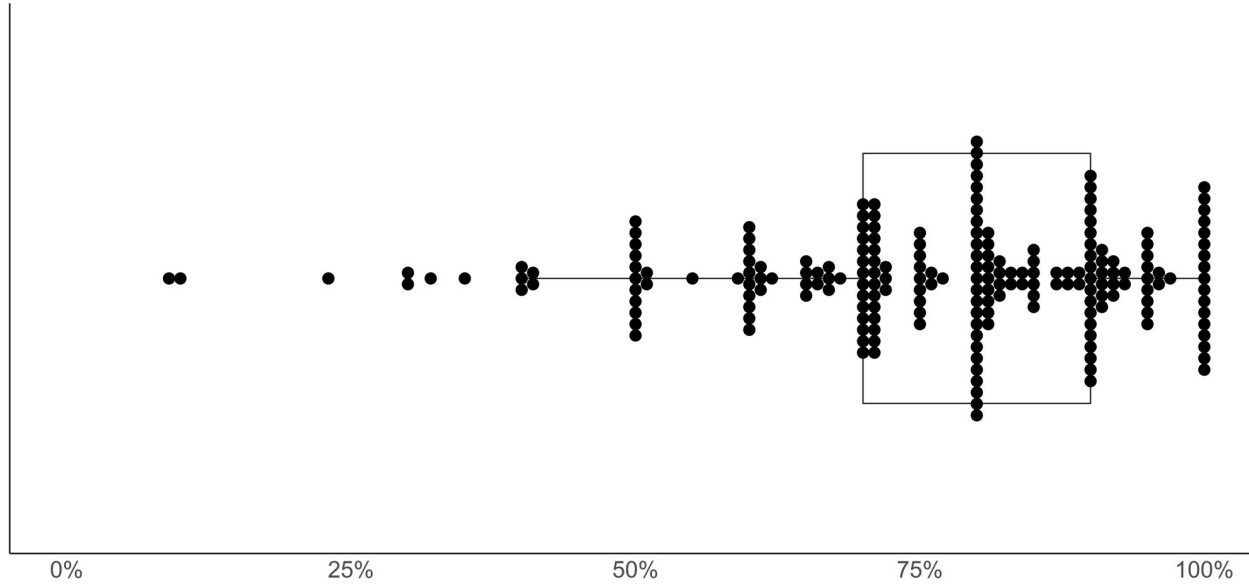

**Fig 10. Animal cognition researchers' self-reported proportion of studies that they have run and then published.** *N* = 208. Each dot represents one researcher's answer to the question "What percent of the studies that you have performed have been published and/or you think will be published?", and a boxplot showing the median and inter-quartile ranges is laid underneath.

the field, from which we identified 13 general themes. These themes are highlighted in Table 6, accompanied by example quotations.

### Replication

We asked researchers what proportion of replication studies they expect would be successful in their area of research, and to what extent their own and other areas of animal

**Table 4. Animal cognition researchers' explanations for why some of their studies go unpublished.**

| Why was your animal cognition study not published? | N | Exemplars |
|---|---|---|
| Inconclusive results | 31 | *"The results were not so clear to discuss something."* |
| Design limitation | 30 | *"Something about experiment was faulty so didn't bother submitting. Or, something was pointed out in peer review that made the study seem not worth trying to publish."* |
| Negative or uninteresting results | 29 | *"It was just too difficult to get "negative results" past referees."* |
| Lack of resources | 29 | *"Mostly [due to] time to write the studies up. I have too many on my "to do" list"* |
| Too few data | 28 | *"Not enough data for reliable conclusions"* |
| Unreliable data | 17 | *"Because the design was weak, the experimenter unexperienced. . . I just wasn't sure whether to trust the data and I did not want to publish any potential false positive/negative findings"* |
| Reviewer bias | 13 | *"Theoretical rivals killed the publication because the outcomes didn't fit with their theory"* |
| Irrelevant data | 8 | *"The data were incomprehensible, and it appeared the animals failed to learn anything related to the task."* |
| Training failure | 5 | *"I often decide not to publish studies if the animals were unable to train to the basic level required to complete the study"* |
| Replication studies | 2 | *"Non-significant results or that only previously published outcomes were replicated"* |

Table 5. Animal cognition researchers' opinions on publication practices in the field.

| Do you have any other comments about publication in animal cognition? | N | Exemplars |
|---|---|---|
| Highlighted the difficulty of getting negative results published | 15 | *"It is next to impossible to publish negative results in animal cognition."* |
| Highlighted the difficulty of interpreting negative results | 7 | *"Studies with negative results often needs additional controls to show it is a true negative; most often animal cognition studies are initially designed to control for that a potentially positive result is a true positive. There are many more ways for something to be negative than to be positive, therefor particular care must be given when publishing such data (negative or no results can often be the result of a bad design)."* |
| Highlighted an excessive focus on publishing "exciting" or "novel" results | 7 | *"There are still strong incentives towards publishing "wow!" findings showcasing supposedly "clever" or "human-like" abilities."* |
| Lack of time to publish everything | 3 | *"I just have not had time to publish them."* |
| Other/Other barriers to publishing in animal cognition | 15 | *"There is a constant pull of the wishful thinking. If we let this go on unchecked, it will eventually converge on what people already think is true and/or what they wish were true."* |

cognition would experience a replication crisis, if many of its studies were replicated. If 100 typical studies in their research area were replicated, researchers believed that 65% (IQR: 50% - 75%) would replicate successfully if the replication study tested a new sample of the same size with the same protocol as the original study. If these replication studies used sample sizes of 1000, researchers estimated that 72% would replicate successfully (IQR: 50% - 82%, Fig 13).

Predominantly, researchers somewhat agreed (34.0%) or somewhat disagreed (30.1%) that their area of animal cognition research would experience a replication crisis if attempts to replicate most of its studies were conducted, and they either somewhat (43.7%) or strongly (29.3%) agreed that some other areas of animal cognition research would experience a replication crisis. Researchers tended to somewhat agree (38.0%), or neither agree nor disagree (31.2%) that they could identify which animal cognition studies would successfully replicate and which would not (Fig 14). When asked about the importance and prevalence of replication studies, researchers disagreed (50.7%) or strongly disagreed (20.1%) that enough replication studies were performed in their area of animal cognition research. These Figs were matched when researchers were asked to consider replications in animal cognition research in general (disagree: 55.5%, strongly disagree: 23.6%). The vast majority of researchers agreed (34.8%) or strongly agreed (54.8%) that it is important that replication studies are performed in animal cognition research (Fig 15). We received 64 free-text responses about replication in the field, with researchers most often highlighting various complexities and nuances of replication in animal cognition research Table 7.

## Belief

When reading papers in their own area of research, and other areas of animal cognition research, our sample reported often or sometimes agreeing with the authors' conclusions (own area: often: 58.4%, sometimes: 38.3%; other area: often: 58.5%, sometimes: 36.2%, Fig 16). Researchers somewhat and strongly agreed that their beliefs about animals' cognition are affected by both scientific experiments (strongly agree: 55.7%, somewhat agree: 34.3%) and their day-to-day experience with animals (strongly agree: 31.9%, somewhat agree: 34.3%, Fig 16). When asked to choose between scientific experiments and experience with a slider response (with science at one extreme and experience at the other), researchers tended to say

## When performing a statistical analysis...

**Fig 11. Animal cognition researchers' self-reported confidence in their own statistical analyses, N = 210.**

their beliefs were more driven by science, although a range of responses were observed (median: 31, IQR: 19–51, where 0 is exclusively based on science, and 100 exclusively based on experience, Fig 17). We received 42 free-text responses about beliefs in animal cognition, from which we identified 5 common themes. Table 8 outlines these themes and provides example quotes, and, although it did not fit one of our themes, we highlight another interesting quote below:

*I think you can almost always find a scientific paper to confirm your beliefs, and can find a way of justifying paying attention to that one, and ignoring one that might give different results. I don't mean this cynically—but humans are very good at piecing together a plausible seeming story with limited evidence! (We're good storytellers, and it can take a lot of evidence to dissuade someone from a good story!)*

### Miscellaneous

Throughout the survey, we perceived five themes across our survey blocks that our within-block coding did not identify. As such these themes were not those systematically extracted, but themes we subjectively believed came up across blocks and wanted to highlight. These were the role of theory in animal cognition, the need for an individual-level focus in research, academic incentives, the large amount of heterogeneity across animal cognition research, and the uncertainty surrounding the causes and implications of negative results. We provide representative quotes for each below.

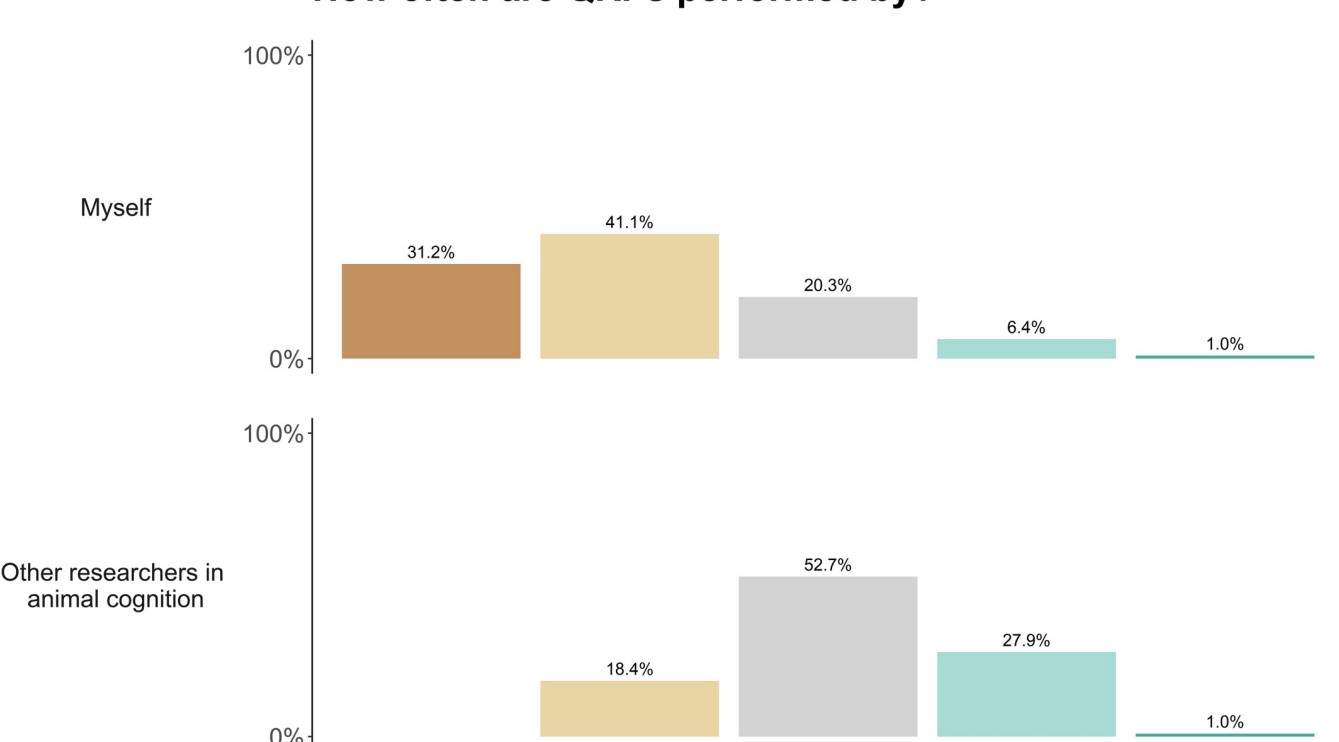

**Fig 12. Animal cognition researchers' self-reported use of questionable research practices, and their estimated use of questionable research practices (QRPs) by other researchers in the field.**

## Theory

"*In my view a far bigger problem is poor theorizing [compared with replication]. A lack of formal theory (as exists in evolutionary biology) combined with "scala naturae" thinking, a lack of consideration of natural history and incentives to show that your study animal is "clever" or human-like are major problems for the field.*"

## Individual-level research

"*It is unfortunate that Single-Case experimental designs (single subject, single-organism, etc) are not used more often, which are known to (a) highlight replication and reproducibility, (b) avoid many hypothesis-testing issues (including, but not limited to those listed above), and (c) avoid many group-design limitations for behavioral research.*"

"*Learning occurs in individual organisms, not aggregate population parameters, yet the field is convinced chasing p-values provides meaningfulness. There's a fundamental misunderstanding of what NHST offers us. Behaviorists could look at a cumulative curve from a single organism, and show an effect because the learning was so obvious. The cognitivist turned to NHST because effects were not clear, so more digging was required because the quantitative imperative required quantitative measurement for an enterprise to be considered science.*

Table 6. Animal cognition researchers' comments about the use of statistics in the field.

| Do you have any comments about statistics in animal cognition? | N | Exemplars |
|---|---|---|
| Lack of training | 15 | *"I think that a lot of statistical mispractice also stems from missing knowledge/proficiency regarding statistical software and/or methods/ tests. The majority of animal cognition researchers have a very sparse statistical education and are therefore self-taught. This can be a huge potential for errors."* |
| Complex statistics as a barrier | 11 | *"The increasing use of highly complex stats (e.g. Bayesian GLIM modelling) doesn't always help. I'm doubtful if most users can work out which variables are being treated as fixed effects in their analyses, for instances, and which of them should be. I certainly can't!"* *"This comment goes beyond animal cognition, but basically you need a second PhD in statistics to handle analyses these days. I think we are all doing our best, but there is only so much we can teach ourselves about statistics when we are also bogged down with our actual research, teaching, grants, etc."* |
| A problem of low power or small sample research | 9 | *"Depending on the species, many problems stem from smaller sample sizes being treated in the same way as large samples."* |
| Incentive structure promotes questionable research practices | 9 | *"I also think current publishing requirements and standards are much to blame for these practices. Papers need to be short and concise, and it is more difficult to write a nice story when unexpected findings were not predicted. Journals also want the newer, exciting results more often than a simple, non-significant story. This doesn't mean these practices are acceptable or should be done, but not everyone can/will fight for ethical scientific standards when an easier solution is highly rewarded."* |
| Questionable research practices not necessarily bad | 8 | *"Sometimes it does happen that you conduct a study with very different intentions than the result you get. In hindsight it would have been a reasonable prediction, and framing it as such can help make a paper clearer." "In some cases I don't think there is anything wrong with this but there is a fine line."* |
| Discussed solutions | 7 | *"I worry about trendy "bandwagons" and fashions. Rather than prescribing particular approaches (e.g. we should all be Bayesians now) statistics should be reported clearly, transparently and in detail (e.g. I have no issue with people reporting p values if they want to, as long as they report effect sizes, associated errors and confidence intervals and visual representations of raw data)."* |
| An anecdote of QRPs | 6 | *"I have occasionally heard researchers pushing for collecting additional data to boost a trend, but it is difficult to estimate how common this practice is."* |
| Individual-level statistics important | 5 | *"At times, the search for population statistics obscures the attempts to understand individual variation. In other words, trying to forge a coherent analysis of many small NS may be more fruitful than statistics based on one large N."* |
| Collaborating with statisticians | 5 | *"I find myself not so knowledgeable about new statistical techniques, so use a statistician."* |

Four themes with a smaller number of responses are not shown (pre-registraion as a named solution (3), Bayesian statistics as a named solution (3), QRPs being less of a problem in animal cognition research (2), and the dangers of dichotomising research at $p = 0.05$ (2).

*NHST poisoned the well, and is directly to blame for the first 3, and 5th, bullet of your list above* [said in reference to the questionable research practices of: performing many analyses and selectively reporting the statistically significant ones; data dredging/p-hacking/fishing for significance and; collecting more data until a significant/desired result is obtained]."

"*[In respect to replication] This approach assumes that the observer will have no effect, and that variables that will affect the study but that may not be mentioned it (because there are an*

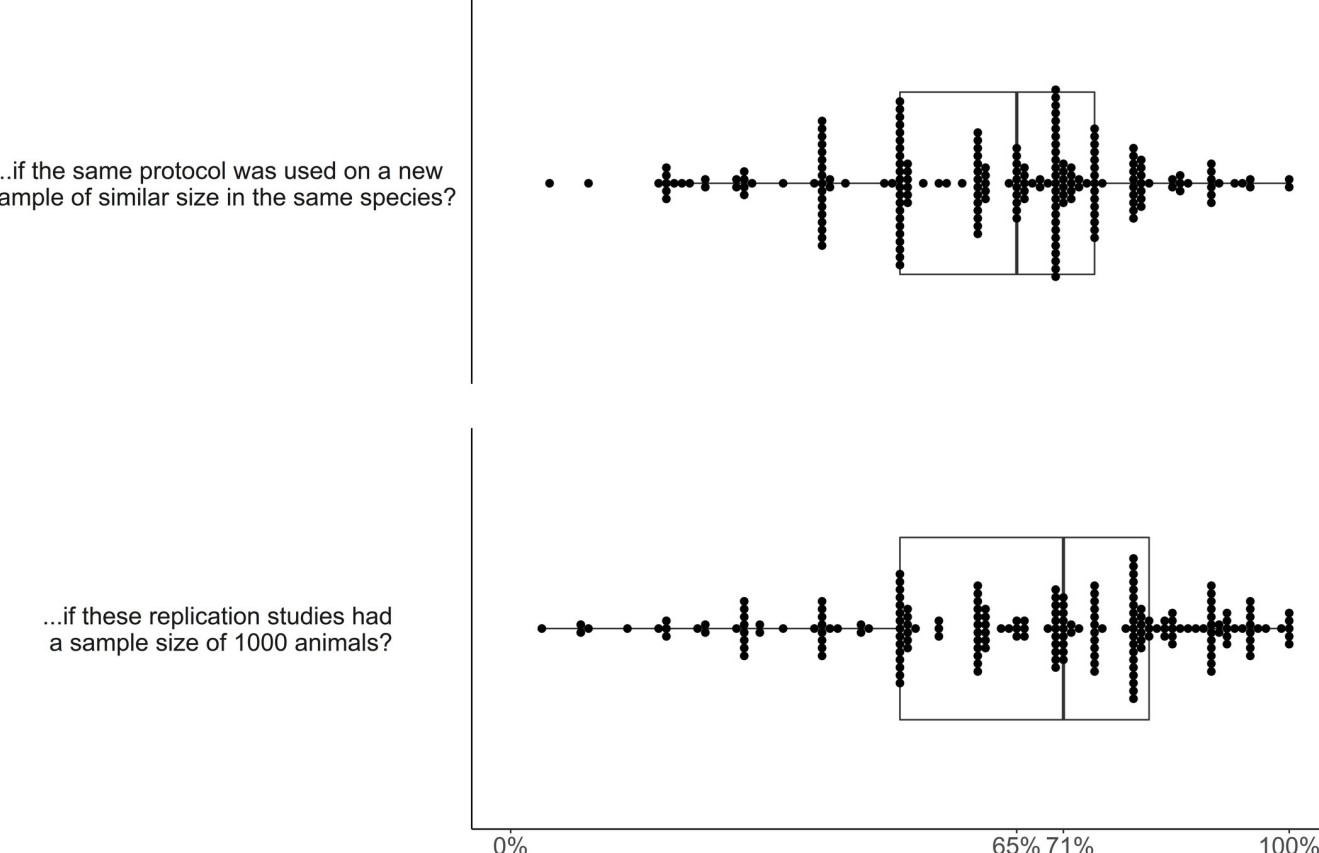

**Fig 13. Animal cognition researchers' predictions of replication success in their field, $N_{same\ sample\ size}$ = 207, $N_{large\ sample}$ = 205.** Each dot represents one researcher's estimate of what proportion of studies in their research area would successfully replicate if these studies were replicated with a similar sample size (top panel) or a sample size of 1000 animals (bottom panel), with boxplots showing the median and inter-quartile ranges laid underneath.

infinite number of variables in even the simplest study that all of them cannot be mentioned and/or equated). This approach assumed that the history of the nonhumans will be the same in replication, which can never be assured and is all too often ignored. This assumes that such things as "personality variables" will all be "smoothed out the greater the N. This assumes that the "observer" is independent of the date collected and that the observer's actions do not differentially affect the actions/behavior of the organism observed. Such assumptions can't possibly be valid for living beings with individually distinct histories—in anything like the way that are true for chemistry and physics."

## Negative results

"*I usually never finish a study which I realize was misconstrued when I see the first behaviors of the animals. Oftentimes it is easy to arm chair-design a study which turns out to be impossible for practical and other reasons. This is not saying that I have not published finished studies with negative result. However, studies with negative results often needs additional controls to*

## A Replication Crisis?

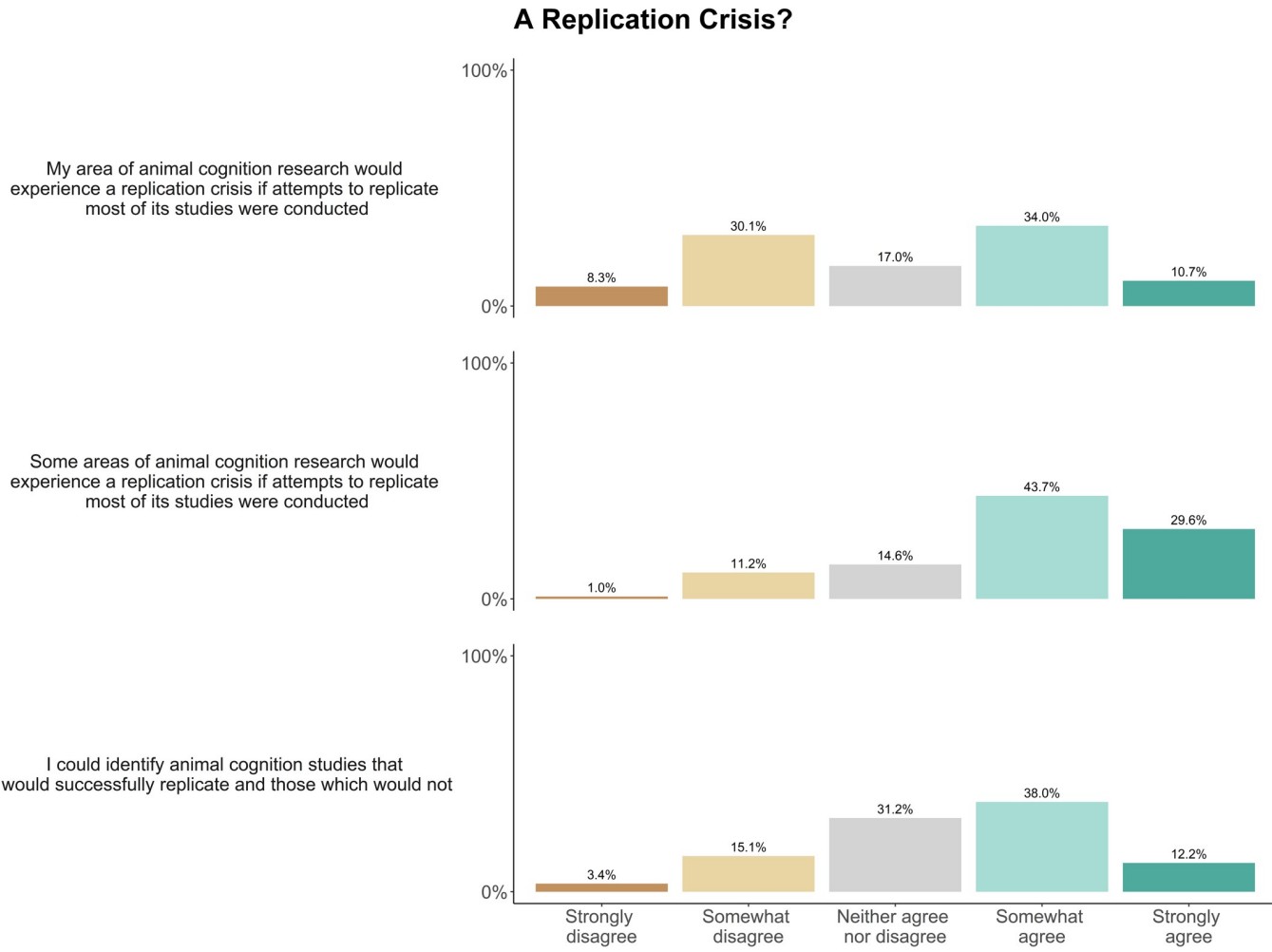

**Fig 14. Animal cognition researchers' perceptions of a replication crisis in the discipline, and their ability to identity studies that would not replicate, N = 210.**

show it is a true negative; most often animal cognition studies are initially designed to control for that a potentially positive result is a true positive. There are many more ways for something to be negative than to be positive, therefor particular care must be given when publishing such data (negative or no results can often be the result of a bad design)."

## Incentives

"*In my opinion, the drive to publish 'exciting results' is driven by the expectations of funding bodies, and the general competitiveness of the academic system, that expect people to constantly produce ground-breaking new research. Not all research is or can be ground-breaking, but is a necessary part of research progress, such as proposals for methodological improvements. Such research deserves more support from the research community. Shifting the weight in expectations on researchers might reduce peoples' need to over-interpret borderline p values and report 'impactful' findings where there is really little to none.*"

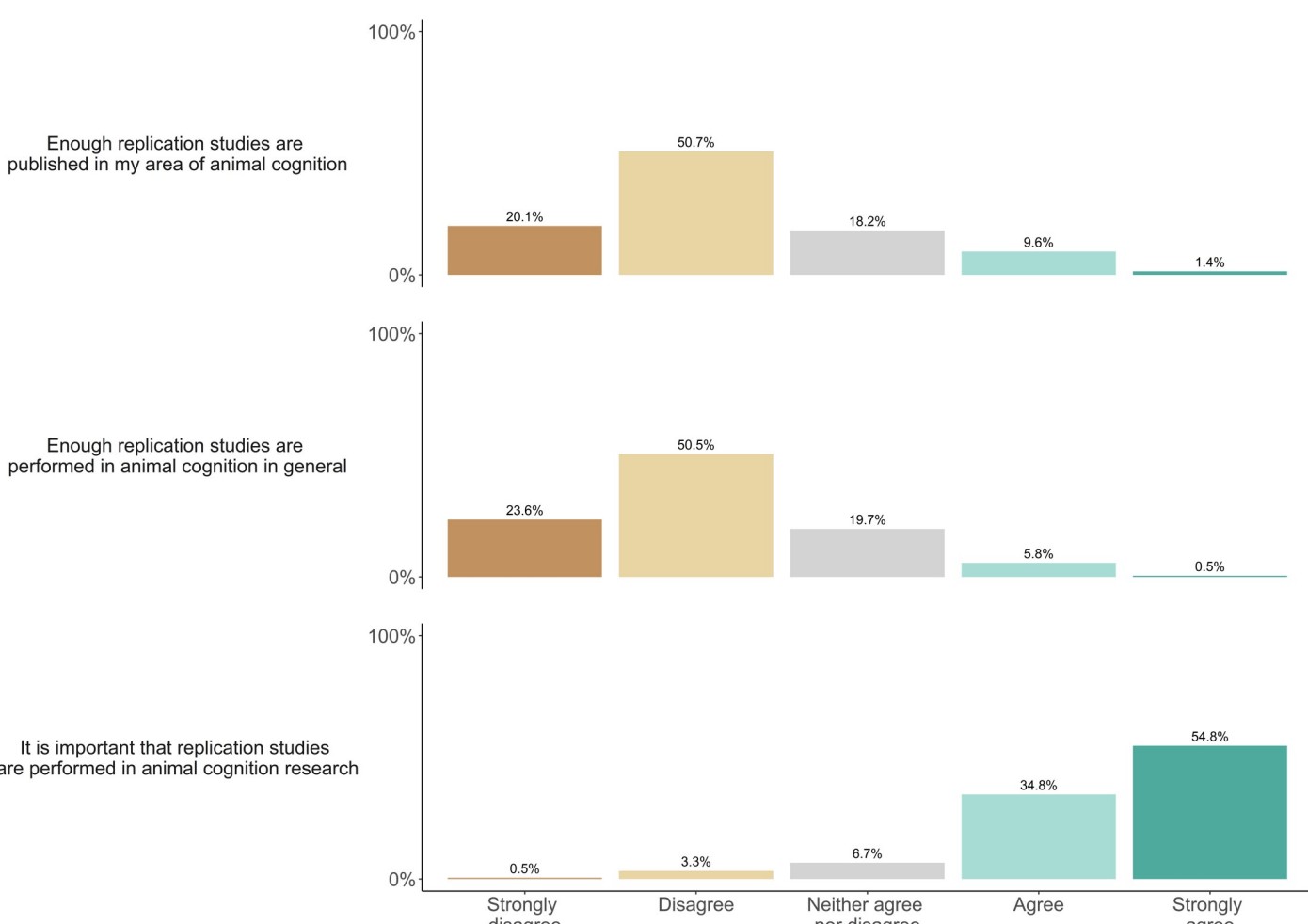

**Fig 15. Animal cognition researchers' perceptions of the frequency and importance of replication studies in the discipline, N = 210.**

**Heterogeneity.** The issue of heterogeneity is perhaps the largest caveat to our survey results. For many questions, some researchers said that their answers would vary depending on the exact area of research or identity of the researchers. For example, researchers may believe that questionable research practices are rarely used by most researchers, but often used by a minority, or believe that results in one area of research (e.g., animal learning) might be more replicable than others:

> "*I wish the second section above had used the terms of the first section (i.e., rarely, sometimes, always. . .) because in my experience the biases occur with some authors/scientists rather than in a specific section of animal cognition. I also wish the statements in the second section hadn't included the word "strongly" in them. There are certain authors whose papers I can predict will have questionable methods and over-interpreted results rather than finding that in a specific area of animal cognition. Generally, I find more careful work in comparative psychologists' papers than in papers from other fields for animal cognition work.*"

**Table 7. Animal cognition researchers' beliefs about replication in animal cognition research.**

| Do you have any comments about replication in animal cognition? | N | Quotes |
|---|---|---|
| Complexity of replication | 27 | *"As results appear to be heavily influenced by all sorts of things— history and experiences of the animals, particularities of the facility, particularities of the group structure and (sub-)culture of the animals, the experiment paradigm details, reward distribution, training protocol. . .—replication is horribly difficult."* |
| Lack of Incentives | 16 | *"Journals are more and more looking for novel ideas and results, and unfortunately replication studies are seen as unimportant, unless they shockingly dismiss some big ideas."* <br> *"Like other areas of research, the current publishing system values novelty and I believe this to be a major limitation that has discouraged replication in cognition research."* <br> *"Funding to conduct replication studies is more difficult to obtain, than for novel studies"* |
| Importance of converging evidence | 8 | *"The term "replication" is not entirely straightforward. A strong replication is not always using the same protocol or the same stats and arrive at the same conclusion. I believe that a well-replicated result is something that shows to be correct when using a variety of methods and approaches and still arrive at a very similar conclusion. I think that this is true for several areas in animal cognition."* <br> *"I think interpretation is a bigger issue than replication. Even under describing the methods (sometimes many important details are omitted) is a bigger issue. I DO think replication is important, but as we look across experiments, even though they are not exact replications, I think we can see the trends in what is likely a real effect and what may be something that could not be replicated. We should be training students how to look for these trends though."* <br> *"Depends very much on the topic, and what is meant by replication. In controversial areas, such as episodic memory in animals, there has been numerous attempts to demonstrate or refute, but often with different species. This involves attempts to replicate a phenomenon, but not necessarily a particular study."* |
| Issues with bias and validity more problematic than replicability issues | 7 | *"At least in my subfield of animal cognition the replicability of studies might actually be viewed as a negative because the assumptions and interpretations of the studies are fundamentally flawed. So the studies replicate, but researchers take those replications as additional evidence for the validity of their paradigm when it is not."* |
| Between area heterogeneity in replicability | 5 | *"Confidence in my own area of research has to do with the common practice of "embedded replications" of successful previous work in my work. My lack of confidence in some other cases has to do with demonstrations with sparse background literature/experimentation to back it up."* |
| Legal or ethical barriers to replication | 3 | *"One problem with replication in the context of animal work is the clash with the ethical pressure to minimise animal use."* |

*"I don't agree with the question about my belief about other researchers tending to make weak or strong claims given their data. The answer should have been one of choosing the percent that I believe make stronger claims than warranted. Most researchers (70%) I believe make appropriate claims, but some (30%) do make stronger claims than warranted."*

*"[Some of the] responses above are misleading averages. Sometimes I find on peer review I have overclaimed, sometimes that I have underclaimed, and the same is true of authors whose work I review."*

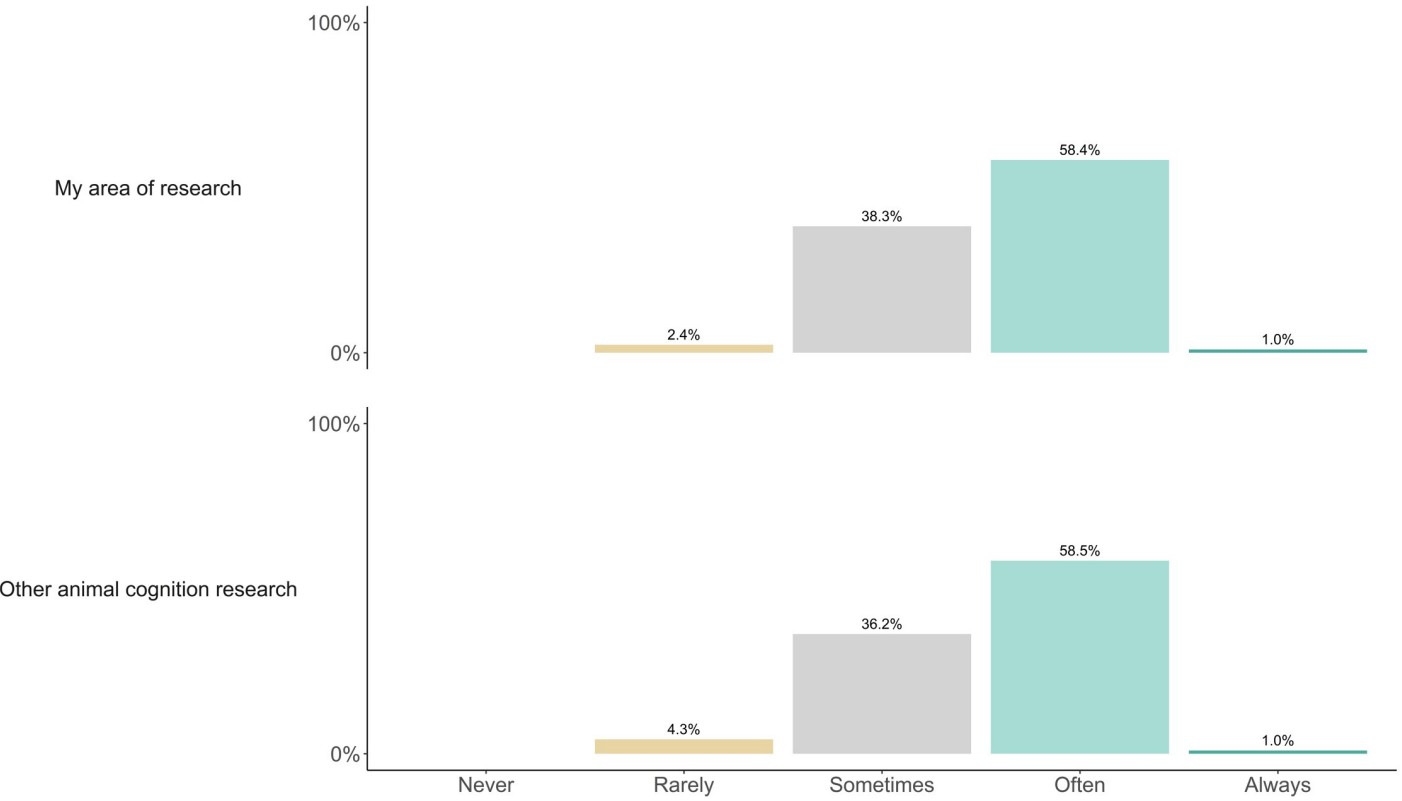

**Fig 16. Animal cognition researchers' tendency to agree with the conclusions of papers in their own and other areas of research.** N = 210.

## Discussion

Our survey provides a picture of animal cognition researchers' beliefs about bias and scientific practice. From 1001 invitations, we received 210 completed surveys, from which we analysed data on a range of controversial topics and possible biases in animal cognition research. While it is likely that there was a self-selection bias in who completed our surveys, with researchers who have stronger feelings about bias in the field presumably being most likely to complete our survey, 210 completed surveys reflects a large number of recently active animal cognition researchers. Before discussing the individual survey topics, we wish to outline what we believe the data from surveys like our own are useful for and what they are not. Specifically, we do not believe that these data are very accurate or representative data of all animal cognition research-ers' beliefs, or very accurate estimates of, for example, the absolute rate of questionable research practice use in the field (see e.g. [42]). Rather, they must be interpreted considering the likely sampling biases in who participated in our survey and how their answers were lim-ited by the way the questions were asked. Specifically, the strongest sampling bias is likely that the researchers who completed the survey, and especially those providing detailed free-text responses. These individuals are likely those who have thought most about some of the issues presented in the survey, and are potentially the most concerned about some of these issues (e.g., reliability) than researchers who did not complete the survey. This might mean that some of the quantitative estimates, e.g., perceptions of a replication crisis, might overestimate the "average" response of animal cognition researchers to this question, but equally might

**My beliefs about the cognition of animals are affected by...**

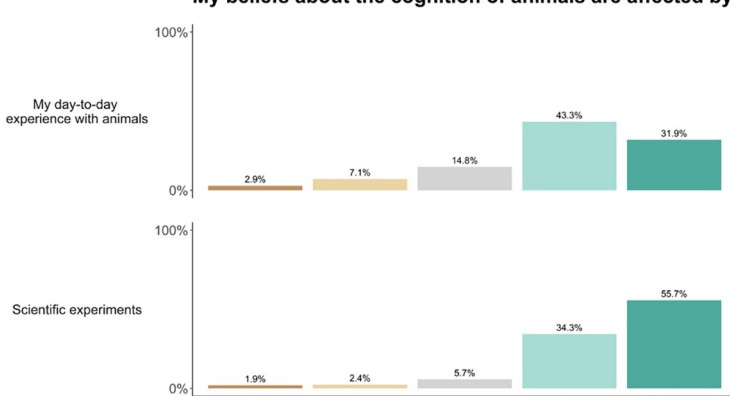

**My beliefs about the cognition of animals are determined...**

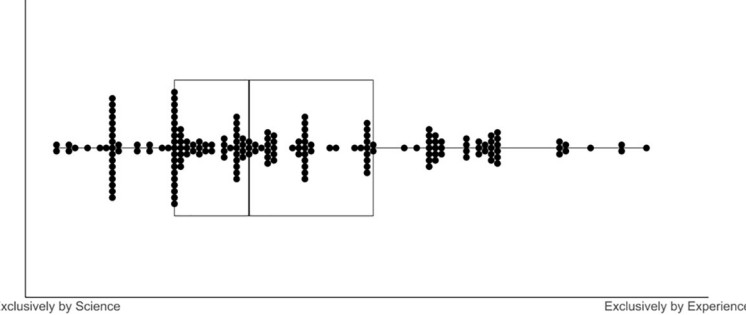

**Fig 17. Animal cognition researchers' reports of the role of science and daily experience in shaping their beliefs about animals' cognition.** N = 210. Top: Answers to individual questions on the role of experience and science. Bottom: Researchers' responses about the relative role of science and experience. Each dot represents one researcher's response. with boxplots showing the median and inter-quartile ranges laid underneath.

underestimate the concern about bias within their own results–if these researchers are more likely to e.g., adopt blinding strategies. Nevertheless, each individual response that we received reflects the opinion of a particular animal cognition researcher, and thus are inherently meaningful pieces of data, with detailed full-text responses available at osf.io/6j7kp. For individuals new to the field, for example new PhD students, the data offer an accessible window into some of the perceived issues within animal cognition research, and the commonality of some of them, that are often not readily available in the literature in such a candid fashion. Moreover, these data can provide evidence of publication bias, questionable research practices and (lack of) confidence in some of the field's findings, and the mechanisms underlying them, which can be used to both stimulate debate within research groups and support theoretical arguments about the status of animal cognition research. Finally, the data–especially the free text data— offer a clear window on the barriers researchers feel inhibit progress in animal cognition research. These data will be particularly useful for PIs, editorial boards, hiring committees and funders to make decisions on policy changes that might facilitate stronger science in animal cognition. We now discuss the specific findings of our survey, and compare these to similar studies across disciplines, before outlining some of the ways we believe animal cognition research can improve in light of these data.

**Table 8. Animal cognition researchers' beliefs about the role of science and day-to-day experience in shaping their beliefs about the cognition of animals.**

| Do you have any comments about belief in animal cognition? | N | Quotes |
|---|---|---|
| Experience acts as a source of scientific hypotheses | 11 | *"Observations and experiences may give you hints about possible study questions. They leave you with impressions of animals' mind that require further digging into. Science, however, is absolutely vital to yield actual knowledge."* |
| Bias in scientific results prevents them impacting beliefs | 10 | *"I am hesitant to say that the results of scientific experiments affects my beliefs about animal cognition. This is mainly because I know that many studies are poorly executed, and it is the norm to make huge claims with no or limited data to back it up. Certain authors make careers out of their great skills at hyperbole, and I find this ethically unacceptable. On the other hand, there are authors that perform good science and don't make exaggerated claims. These studies I take seriously, and the work of such authors does indeed have the potential to affect my beliefs about animal cognition."* |
| Science can answer questions experience can't | 9 | *"There are a lot of species studied. . .so even experts in the field could only obtain knowledge about that species by reading papers (for the most part)."* |
| The role of science and experience shaping belief varies depending on the topic | 7 | *"There are deep questions and shallow questions. Deep questions, like whether a crow has consciousness, can only be answered by a scientific theory of the concept. Shallow questions, like whether a dog has a memory of where a bone was buried, can be answered with empirical observations."* |
| Experience with animals can be valid data in itself, and/or necessary for producing valid data | 5 | *"What exists in the literature is relatively limited compared to the richness of experiences that working with animals regularly offers. Individual experiences, even if one-offs, can be very provocative indicators of cognitive potentials."*<br>*"The definition of "cognition" is fuzzy and not recognized as "fuzzy," most students are taught to "operationalize" and to "standardize" their data, before they know enough about the natural behavior of the animals to be able to perform those types of procedures appropriately and it is in these procedures that bias inevitably and unwittingly enters into their research."* |

## Bias

Overall, researchers were wary of bias across animal cognition research. Researchers often agreed, or neither agreed nor disagreed, that the results and theories across animal cognition are strongly affected by researchers' biases. For example, some researchers' qualitative responses suggested that they believe bias not to be uniform across the field, instead reporting that certain topics and researchers may be more likely to be affected by bias than others. Similar to other survey studies of scientific bias, our participants were generally more concerned about bias in others' research than their own [43,44], although there were exceptions, often being both very conscious about the possibility of bias in their own and others' work. This was especially pronounced for experimenter bias, where researchers did not appear especially concerned that they might be biasing their own results, and were, on average, confident they could perform fair tests of animal cognition. This somewhat conflicts with primary data suggesting that experimenter effects can have a large influence on animal behaviour [50,51], and that blinding procedures are rarely reported [19]. This confidence in avoiding experimenter effects might reflect an overrepresentation of researchers in our survey who take steps such as blinding to minimise these effects in their research, or who believe their experiments should be unaffected (e.g., by not being in contact with animals during testing due to using touchscreen apparatus). However, we also received some strong responses from researchers who fervently

believed that researchers always hope for particular results and thus should always be concerned that they might be biasing their results, and several researchers noted how bias can be embedded in research programmes even before data collection begins.

Similarly, while researchers believed that other animal cognition researchers sometimes use questionable research practices and overclaim when submitting papers to journals, they reported that they themselves were less likely to do so. This replicates the patterns observed in similar survey studies in psychology and ecology and evolution [43,44]. However, several researchers caveated their answers in the free text responses, highlighting how bias might not be uniform across the field. For example, some researchers reported that some areas of research might be weakly affected by bias and questionable research practices, but other areas and researchers more so.

Our survey results also provide direct evidence of publication bias in animal cognition research, self-reported by active researchers in the field. The median percentage of studies researchers reported publishing was 80%, although over 10% researchers reported publishing less than 50% of their studies. These figures may underestimate the prevalence of publication bias both within our sample and in animal cognition more generally. Within our sample, the figures may be an underestimate as published findings are likely easier to recall for participants while they were completing the survey (i.e., an availability bias [52]). In animal cognition more broadly, the figures may be an underestimate if our participants were more likely to publish negative results than the average animal cognition researcher. While researchers reported a journal or reviewer enforced publication bias against negative results or against results not in line with "preferred" theories, many researchers also reported not attempting to publish studies with difficult to interpret results, or those that had flaws in the experimental design or were otherwise perceived to be low quality. Notably, this decision not to publish was often the researcher's own, with a lack of time or incentives often cited as the limiting factor. Combining participants' quantitative and qualitative responses suggests that across most areas of animal cognition research, many studies have been performed but not published. This suggests that the published literature may not be representative of all research conducted in animal cognition, which makes it hard to evaluate the strength of evidence for many effects from the literature alone. Because of this, attempts at evidence synthesis, whether through meta-analysis, review articles or even introductions to experimental pieces should seek to evaluate the extent and consequences of publication bias in their topic area.

Given the degree of concern about bias in research in animal cognition–especially in others' research–scientists in animal cognition could take steps to mitigate bias, and, through embracing transparency throughout the research process, demonstrate this trustworthiness to others. While there is currently no central repository or systematic method for study registration (c.f. https://clinicaltrials.gov/ for medical trials), research groups could seek to publicly archive all studies they conduct, which would allow other researchers to assess the strength of evidence not just from individual studies, but in relation to the entire research programme they have come from. Within individual studies, registered reports in which authors receive peer review and in-principal acceptance before conducting data collection [53,54], have the threefold benefit of removing results-dependent publication bias, pressures for certain results during data collection, and the ability to strengthen study design prior to data collection. Finally, effective blinding procedures should continue to be used where possible, during both data collection and during inter-rater reliability procedures. Where blinding cannot be performed, researchers may wish to introduce heterogeneity into their study designs–for example by using many different experimenters, in order to attempt to quantify any experimenter effects.

## Morgan's canon

Over 70% of our sample somewhat or strongly agreed that Morgan's canon is important to use when interpreting the results of animal cognition experiments. Superficially, this contrasts with a large body of literature criticising the canon on the grounds that there is no reason to privilege "simpler" or "lower" explanations of animal cognition over more "complicated" or "higher" explanations [20,21,55–60]. However, participants qualitative responses revealed a more nuanced picture: Many of those who also provided free-text responses, a) recognised the inherent ambiguity and multiple interpretations of Morgan's canon, and, b) cautioned against a blind application of Morgan's canon. Of those who defended the canon, most defended a particular principle associated with it (e.g., parsimony and phylogeny), rather than the canon itself. Evidently, Morgan's canon and related concepts elicit a plurality of opinions. Because of the variety of interpretations and justifications for invoking the canon, or e.g., parsimony, arguments should not likely be evaluated based on the authority of these principles alone–because researchers might understand them differently. Rather, researchers should strive to make the assumptions and justifications for favouring one hypothesis over another explicitly–something that could be achieved through formal modelling (although, see "Theory and modelling" section in discussion).

## Replication

Over 70% of our sample agreed or strongly agreed that some areas of animal cognition could experience a replication crisis, and, in our sample, slightly more researchers agreed (44.7%) than disagreed (38.4%) that their own area of research would experience a replication crisis, if attempts to replicate its studies were performed. This suggests a large degree of skepticism about the robustness of research findings in some areas of animal cognition research, or of the ability of replication studies to repeatedly identify certain effects. However, such skepticism is common across sciences, with 52% of 1576 researchers surveyed across fields including biology, chemistry and physics, reporting that there was a "significant" reproducibility crisis in their field [61].

In our survey, researchers near unanimously agreed that replications were important, and not performed frequently enough (Fig 15), mirroring the view of ecology and evolution researchers [41]. A smaller number of researchers noted that replication studies may be less important than seeking convergent evidence of phenomena. These views echo wider discussions about the role of direct and conceptual replications in psychology, with conceptual replications being essential to provide robust evidence of general psychological effects (see e.g., [62]). However, an exclusive focus on conceptual replication can be problematic when it co-exists with a publication bias against negative results (see e.g. [63]), as "converging evidence" for spurious effects can populate the literature [48]. Hence, if the rate of false discovery is or has been high in animal cognition research, a short-term focus on direct replication may be necessary to identify those effects that are locally robust and those that are not (note, however, that the direct vs conceptual distinction in replications is a false dichotomy, and see [64] and [65] for perhaps more useful classifications of replication, and [66] and [63] for applications to animal cognition).

That areas of animal cognition research might experience a replication crisis, combined with the general belief that replication studies are not performed often enough, is a finding similar to those in other fields [41,61]. However, unlike many other fields, the possibility for independent replication is low for most questions in most species. This means that it is critical for individual labs to assess the likely robustness of their own findings, and these survey data can provide a starting point for such discussions. In the interim, researchers may wish to be

cautious when citing and reviewing animal cognition research that they believe shows some hallmarks of irreproducibility (see [27,67] for discussion of this).

## Belief

Researchers reported that their beliefs about animal cognition are influenced by both the results of scientific experiments and their own personal experience with animals. Typically, researchers viewed science and experience as synergistic, with experience often cited as the source of scientific hypotheses, and necessary for designing good experiments. A smaller number of researchers also endorsed every-day knowledge as a valid source of data that could be seen as equally strong as some scientific data [68], although researchers often noted that the role of science and experience depended on the question at hand–there are some, often trivial, questions that can be answered readily through experience, yet many researchers reported that some knowledge can only be accessed through systematic scientific study. Finally, researchers noted that for many species that they have no experience, rely on the scientific literature to form their beliefs, which requires them to trust the findings of their colleagues.

## Miscellaneous

While our survey focused on five blocks of questions that we were particularly interested in, oftentimes researchers' free-text responses went beyond these questions and highlighted specific issues that were not directly solicited by the survey. For example, a researcher offering reservations about the press coverage of animal cognition research, or species biases in what is tested and interpreted, as well as biases based on the location of where research is conducted. We encourage the reader to view the full database of open-text responses to make the most use of these low-frequency data from this survey (osf.io/6j7kp). However, there were five themes that we interpreted that went beyond our initial survey aims. These were theory, individual-level research, incentives, heterogeneity and interpreting negative results. Each of these topics should be key discussion points concerning how animal cognition research should progress, some which can be applied readily (e.g., focusing on individual-level research), and others that are longer-term issues (e.g., the role of theory).

**Theory and modelling.** A lack of theory may impair a field's progress. Without strong theoretical grounding, research programmes may fall into a process of testing vague, verbal hypotheses that are only loosely connected to the data the experimenters collect, and this data (and the verbal hypothesis) can be interpreted in almost any way the researcher chooses. In animal cognition, this might result in research programmes that continually use hypothesis testing within single studies to make large claims, such as e.g., confirming an animal is "clever" or possess human-like (or any other target animal) abilities [69]. In contrast, formal theories, be they logical, computational or mathematical, can have a string of benefits. For example, they might increase the precision and communication of hypotheses, make clear predictions, and offer the ability to simulate effects (see [70–76] for discussion). In animal cognition research, evolutionary theory [77], and learning theory [78,79], are two possible sources of strong theory to ground research programmes in, and tools and tutorials for using theories like this in study design and analysis are increasingly available [80,81]. However, it is unclear the extent to which formal models can effectively be generated for all research lines of interest. This uncertainty can be illustrated on the example of mirror recognition studies. Clearly, how animals respond to their reflection in a mirror is an interesting question, and one that can be interpreted in relation to evolutionary and learning theory [82–84]. However, just how much formal modelling can bring to studies of mirror recognition is unclear. It seems reasonable that, at first, the primary focus should be on collecting high quality data and discovering robust

statistical effects, from which theories could be built. For many questions in animal cognition, especially those where animals are not under a large degree of control, high quality documenting and description of behaviour is likely to present a key step in any research programme. Nevertheless, the role that more formal theory and modelling should play across animal cognition research is a complicated issue, and one that merits further specific discussion within individual research programmes in animal cognition.

**Individual level research.** Related to the concern about generating high quality data is the question of what should be the focus of animal cognition research: the individual animal or the average response of a population of animals? Given that psychological effects, occur within individual animals, a clear case can be made that researchers should design their experiments, where possible, with the statistical power to detect meaningful effects within individual animals [85]. This has the twofold benefit of increasing the reliability of research findings (high power at the individual level entails high power at the group level), but also of being able to quantify and describe meaningful individual differences in behaviour [7,86].

**Negative results.** Throughout the survey, researchers often returned to the issue of negative results. They both remarked that they are hard to publish due to journals rejecting them, but also hard to interpret, due to the multiplicity of reasons of why an animal might 'fail' a task or not display a certain effect. This has received previous attention in the animal cognition literature [87,88], and more widely in psychology [89], but with no clear consensus on the way forward. For evidence interpreting positive results to be interpreted effectively, the body of research from which that positive result emerged must be known. In this sense, publishing negative results is essential for meta-analysis and evidence synthesis. However, individually, these negative results are undeniably difficult to interpret, and for this reason researchers must be cautious not to over-interpret the meaning of null results (see e.g., [90]). Similar to mitigating bias, registered reports and study registration seem promising avenues to mitigate publication bias and for labs to document which studies they have performed.

**Heterogeneity.** Across all blocks of our survey, researchers highlighted that their responses would differ for different researchers and areas of animal cognition research. Animal cognition research covers a large range of topics, in a large range of species, by a large range of researchers using many different approaches. Even within individual researchers, it is likely that some results are more affected by their biases than others, and this makes detecting bias and quantitative evidence synthesis difficult in animal cognition. In relation to our survey data, it seems clear that many of our respondents were concerned about several aspects of how animal cognition research is practiced. However, the extent to which this general concern can readily be linked to specific studies or areas of the literature is unclear. One possible approach to address this is to increase the amount of systematic secondary data analysis in meta-research projects that extract data about the research designs, methods and evidence of published findings in certain research themes [91].

**Incentives and improving animal cognition research.** Many of the issues highlighted by our respondents seemed united by the premise that the current academic climate does not incentivise best scientific practice [92]. This is a well-established theme in the broader scientific literature coming out of the "replication crisis" [93,94], and initiatives already developed outside of animal cognition research could help researchers respond to the issues highlighted in this study. As previously mentioned, registered reports and study registration offer a strong method to combat publication bias. Pre-print servers can also facilitate researchers publishing data and claims without or prior to peer-review without the possibility of reviewer bias.

However, while individual researchers and laboratories can make some changes to their research process, the strongest changes will inevitably occur through top-down initiatives. For example, one survey study found that encouragement from journals, institutions and funders

would be an effective method of increasing data sharing rates in psychology researchers [95]. Journal policy changes, for example towards accepting replication studies, registered reports, and embracing more sophisticated standards of evidence evaluating than just statistical significance [91], will be key in motivating researchers to produce stronger research reports. However, ultimately, the degree to which scientific funding and employment structures promote poor quality science must be examined. Although beyond the scope of this paper, many researchers have suggested that current grant culture and precarious contracts, coupled with a strong focus on research output with dubious metrics–such as citation rate and impact factor–are promoting poor research across scientific fields (e.g. [92–94]). Initiatives to combat these issues are gaining traction, such as the Declaration on Research Assessment (DORA: https://sfdora.org/), and, as with any culture shift, there will be a degree of inertia in how fast research organizations can adapt.

## Conclusions

This survey provides a snapshot of animal cognition scientists' beliefs about bias, replicability and practices in animal cognition research. Animal cognition scientists predicted replicability issues in the field and were generally wary of a range of biases affecting the research process, although more so in others' work than their own. On average, they believed questionable research practices and overclaiming to be somewhat prevalent in the research field. The survey provided direct evidence for a publication bias affecting the field: researchers self-reported publishing a median of 80% of their studies, however, there was a considerable variation in their responses. Publication bias seemed to be against negative, difficult to interpret or poorly designed research, and was both reported as self-enforced (i.e., the article was never written or submitted), and journal enforced. Researchers also perceived a journal- and reviewer-enforced publication bias against results contra to established theories and reviewers' preferences. On the whole, our participants displayed a range of opinions concerning bias and replicability, largely mirroring the debates of the wider scientific community when considering reliability of scientific results. These views included advocating for incentive reform and replications, and improving statistical inference, but also stressing the importance of developing theory and seeking converging evidence for theories.

## Supporting information

**S1 File. Further demographic information on the survey participants including a k-means cluster analysis of participants' endorsement of key words describing their research.** (PDF)

## Acknowledgments

We would like to express our extreme gratitude to the 210 researchers who completed our survey and who often provided thoughtful and detailed responses to our questions, as well as constructive feedback. We would also like to thank Elijah Garcia, Francesca Cornero, Edward Legg, and Katharina Brecht for helpful comments on earlier drafts of the manuscript. Finally, we would like to thank the editor and two anonymous referees for helpful feedback throughout.

## Author Contributions

**Conceptualization:** Benjamin G. Farrar, Ljerka Ostojić, Nicola S. Clayton.

**Data curation:** Benjamin G. Farrar.

**Formal analysis:** Benjamin G. Farrar.

**Investigation:** Benjamin G. Farrar.

**Methodology:** Benjamin G. Farrar, Ljerka Ostojić.

**Supervision:** Ljerka Ostojić, Nicola S. Clayton.

**Visualization:** Benjamin G. Farrar.

**Writing – original draft:** Benjamin G. Farrar.

**Writing – review & editing:** Benjamin G. Farrar, Ljerka Ostojić, Nicola S. Clayton.

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
