## [Decision Letter · Decision Letter 0]

16 Jul 2021

PONE-D-21-11911

The Hidden Side of Animal Cognition Research: Scientists’ Attitudes Toward Bias, Replicability and Scientific Practice

PLOS ONE

Dear Dr. Farrar,

Thank you for submitting your manuscript to PLOS ONE. I sent it to two experts in this area and their feedback appears below. As you can see, both reviewers are positively disposed towards this work and see its potential to make a contribution. They nevertheless provide a series of recommendations aimed at improving it. Thus, after careful consideration, I feel that while your paper has merit it does not yet fully meet PLOS ONE’s publication criteria. I therefore invite you to submit a revised version of the manuscript that addresses the points raised during the review process.

We look forward to receiving your revised manuscript.

Kind regards,

Mark Nielsen, Ph.D.

Academic Editor

PLOS ONE

Journal Requirements:

1. Please ensure that your manuscript meets PLOS ONE's style requirements, including those for file naming. The PLOS ONE style templates can be found athttps://journals.plos.org/plosone/s/file?id=wjVg/PLOSOne_formatting_sample_main_body.pdf and https://journals.plos.org/plosone/s/file?id=ba62/PLOSOne_formatting_sample_title_authors_affiliations.pdf

Reviewers' comments:

Reviewer's Responses to Questions

**Comments to the Author**

1. Is the manuscript technically sound, and do the data support the conclusions?

Reviewer #1: Yes

Reviewer #2: Yes

2. Has the statistical analysis been performed appropriately and rigorously? 

Reviewer #1: N/A

Reviewer #2: Yes

3. Have the authors made all data underlying the findings in their manuscript fully available?

Reviewer #1: Yes

Reviewer #2: Yes

4. Is the manuscript presented in an intelligible fashion and written in standard English?

Reviewer #1: Yes

Reviewer #2: Yes

5. Review Comments to the Author

Reviewer #1: This manuscript reports the results of an online survey about animal cognition researchers’ attitudes towards bias, replicability and scientific practice.

Most researchers have discussions about these topics on a regular basis in their smaller circles of colleagues, so these issues themselves – and the diversity of view points that we encounter- are hardly news to the community. However, I think a quantitative assessment of these attitudes spanning a larger sample of researchers, as provided by the current paper, can lend weight to a discussion of whether there is agreement that the field faces potential problems, where they lay and what could be done about them.

Methods and results are clearly presented and fit with what the authors conclude from the findings. Data was available at the indicated repository. I have no major concerns with this paper.

The free-text exemplars are interesting to read, but sometimes it could be made clearer according to which criteria the reported exemplars were chosen. I would also suggest to mention explicitly on p 10 (somewhere around 217-219) that the full range of answers the authors were allowed to share can be found in the OSF data set.

On p 10-11 the authors report that the coders came up with several categories/common themes for the free text analysis. I wasn’t sure how this relates to the different areas of the questions, i.e. whether the referred to “common themes” span all areas or refer to within-area categorization? Sometimes there are categories with exemplars in tables (e.g. Table 4) and sometimes only exemplars are reported (e.g. Table 3). In this respect, I found the result presentation a bit inconsistent.

I think the authors could make a stronger point in discussing the merit or potential use of their results. In line 534-535 they say the results might be used to guide discussions on how animal cognition research might improve in the future, but who should have these discussions - Researchers, universities, journals? For example, could the paper spark fruitful discussion among researchers with opposing views? Does it have the potential to serve as a reference point to inquire with journals, universities, and funding agencies, what their rejection and/or hiring practices are and whether they think their incentives serve the quality of scientific advances, given concerns of the interviewed researchers (e.g. that many researchers report they feel pressured to produce novel & sexy rather than solid scientific outputs)? Does this differ/align with attitudes in other scientific research fields? What initiatives are already out there trying to improve the situation and how can the current paper inform and complement these movements? When I finished reading the manuscript, I was left wondering: “so what?” For me, some more elaboration was missing on how the paper can help improve the situation and who is its target reader group.

Typos & formatting issues:

L 161: missing “t” in “…and he last 4 questions…”

L 168: replace “I” with “they”?

L 173: remove “.”

L 258-259: misplaced line break

L 280: misplaced line break

L 347: misplaced line break

Resolution quality of all figures on the pdf was quite bad, sometimes barely readable. .Tiff quality was fine.

Fig 1-5: I am not sure the full questionnaire needs to be in the paper. I would maybe give a formatting example in the paper and provide the full questionnaire in ESM.

Reviewer #2: Overall Impression:

In this paper, the authors report a timely study on the beliefs of animal cognition researchers as they relate to research practices and bias in their field. The authors surveyed over 200 researchers and find:

- evidence of publication bias,

- that participants feel that results in their field can be biased (due to questionable research practices), although, researchers views of the field in general are more negative than their views of their own practices.

- participants report the importance of replication although many feel that only 60-70% of findings in their field may replicate.

Overall, this paper provides some interesting and important data on how animal cognition researcher view the state of their field in the context of wider scientific issues (and some more field-specific issues, i.e. Morgan's Canon). The paper is well written, with results clearly presented and I think this paper makes an important contribution to our understanding of how animal cognition research operates.

Minor Comments.

Abstract: Solid abstract covering methods and key results, but missing an overarching conclusion. Putting this evidence together, what do the authors feel the main take-home message is (regarding animal cognition researchers views on these issues?).

Introduction: Solid introduction with appropriate referencing throughout. I wonder if a brief paragraph about the study aims (just before the methodology or description of key survey blocks) would help the reader understand exactly what the authors expect to achieve. It is easy to implicitly parse the aims from the intro (i.e., from line 62: "But how effectively these debates are reaching animal cognition researchers in general, and how they are received, has garnered little attention"), but I think a clearer statement would set up the methodology nicely.

Methods - Free-text analysis: I am not an expert on qualitative methods, but I think the researchers are describing some form of thematic analysis. The procedure used to develop themes seems fairly robust but I think that results from this qualitative analysis should be reported consistently (see comments below regarding tables in results).

Table 1. Percentage might a more meaningful descriptor for both variables (with Ns in brackets perhaps)? I.e. easier to assess distribution across bins at a glance?

Tables (2-7). I am not 100% sure why table 2 is included. It presents a "selection" of some biases that researchers think exist, but how were these examples chosen ? Did they emerge from the most common themes? Are they the clearest and most coherent responses (if so, is this an adequate reason to include them here)? If the point of the table is to merely demonstrate a range of views, that is fine, but a clear description of how these specific views were chosen would be useful. I think Tables 3 and 5 have the same issue, while Tables 4, 6 and 7 are fine (as the method for developing themes has been made clear in the methodology and so the examples make sense). The benefit of reading a summary of themes (e.g. like in table 4) is that readers get an idea of both the range of issues that participants mention of their own accord, as well as the relative frequency of these comments. On the other hand, the quotes summarised in table 2 and 3 are less informative as they are not presented in a systematic fashion that help us pick out themes, and we don't get an idea of how popular these various views are. In short, I feel that the reporting of example text should be consistent and align with the analyses described.

Lines 341- Where the authors report "many researchers also reported" I think an N would be useful (to get a better idea of how this comment generalises to the sample.

Lines 399-403. When comparing outcomes on ordinal scales wouldn't it be useful to include a non-parametric test to demonstrate a different distribution of results? This isn't necessary for most sections, but it would be useful for a statement like this: "Predominantly, researchers somewhat agreed (34.0%) or somewhat disagreed (30.1%) that their area of animal cognition research would experience a replication crisis if attempts to 401 replicate most of its studies were conducted, however they either somewhat (43.7%) or strongly (29.3%) agreed that some other areas of animal cognition research would experience a replication crisis." Here, the "however" is important as there is a suggestion that the different responses on these measures are meaningful - a Wilcoxon (for example) would tell us whether this difference is likely to be due to chance.

Line 436. Why highlight this quote? Again, I am not against using these examples but the rationale should be made clear.

Line 454. Why are these themes presented differently from the earlier ones (i.e. in tables with N)? I think that if these themes are important enough to be mentioned they should be presented in a consistent manner (i.e. the number of participants who mentioned these issues should be mentioned, etc.).

Discussion:

Nice to see the authors address general issues with their methods head-on, but I think it would be worthwhile to discuss specific issues in the discussion too and how these may specifically limit interpretations/conclusions. For example, the authors write: "Rather, they must be interpreted considering the likely sampling biases in who participated in our survey and how their answers were limited by the way the questions were asked." Could these sampling biases and their implications be spelled out for us.

Line 544: Be careful of using qualitative responses as evidence of a "general" beliefs in your participants without providing some qualifiers or justification (e.g. "Importantly, researchers’ qualitative responses suggested that they believe bias not to be uniform across the field, instead reporting that certain topics and researchers may be more likely to be affected by bias than others"). Obviously, the same applies to the use of quantitative results, but this is easier to justify given you are reporting raw proportions.

Lines 597-599: The authors interpret the qualitative data relating to Morgan's canon using two themes (a) recognised the inherent ambiguity and multiple interpretations of Morgan’s canon, and b) cautioned against a blind application of Morgan’s canon), however, these themes are not used in table 3 (see comment above regarding consistency in qualitative analysis).

Discussion - Overall: Good summary of key findings as they relate to the survey questions, and good use of free-response text to qualify (or add nuance) to your interpretation of this data. However, one nice thing about qualitative data is that it can go beyond your questions. I wonder if a section of your discussion could address this - for example, the "miscellaneous themes" extracted in your qualitative analysis only warrant a sentence or two in your discussion, but I feel that these findings help identify the next questions to be put to animal cognition researchers. The conclusion does a good job summarising the main results, but I would like to read about: a) taking the results together what are the take-home messages, b) are researcher practices/beliefs/etc in this field different from other disciplines, c) based upon these results what are the authors recommendations (if any) for the future directions of this type of research, education of researchers to reduce bias, changes to incentives that drive these issues. (in short, I think in the discussion the authors could think bigger)

Very minor issues, typos, etc.:

Line 74: "In the current study, we used surveyed researchers’ attitudes"

Lines 153-158. The presentation order here is different from the order in the intro- probably best to be consistent unless there is a good reason.

Line 161. "he" should be "the"

Line 527-529: The following sentence is a bit long and could benefit from being rephrased: "Specifically, we do not believe that these data are a very accurate or representative data of all animal cognition researchers’ beliefs, or very accurate estimates of, for example, the absolute rate of questionable research practice use in the field (see e.g. 42])."

Line 573/574/577. Should "Figs" be "figures"?

6. PLOS authors have the option to publish the peer review history of their article (what does this mean?). If published, this will include your full peer review and any attached files.

Reviewer #1: No

Reviewer #2: No

---

## [Author Response · Author response to Decision Letter 0]

29 Jul 2021

Please see Response to Reviewers document for the response below formatted. 

Editors comments

Thank you for submitting your manuscript to PLOS ONE. I sent it to two experts in this area and their feedback appears below. As you can see, both reviewers are positively disposed towards this work and see its potential to make a contribution. They nevertheless provide a series of recommendations aimed at improving it. Thus, after careful consideration, I feel that while your paper has merit it does not yet fully meet PLOS ONE’s publication criteria. I therefore invite you to submit a revised version of the manuscript that addresses the points raised during the review process.

Thank-you for soliciting reviews from two expert reviewers. We found the reviews very helpful, and have largely amended the manuscript in-line with their suggestions, which we outline point-by-point below. 

Reviewer 1

This manuscript reports the results of an online survey about animal cognition researchers’ attitudes towards bias, replicability and scientific practice.

Most researchers have discussions about these topics on a regular basis in their smaller circles of colleagues, so these issues themselves – and the diversity of view points that we encounter- are hardly news to the community. However, I think a quantitative assessment of these attitudes spanning a larger sample of researchers, as provided by the current paper, can lend weight to a discussion of whether there is agreement that the field faces potential problems, where they lay and what could be done about them. Methods and results are clearly presented and fit with what the authors conclude from the findings. Data was available at the indicated repository. I have no major concerns with this paper.

We would like to thank Reviewer 1 for taking the time to provide a detailed review and for their positive comments about the project. 

The free-text exemplars are interesting to read, but sometimes it could be made clearer according to which criteria the reported exemplars were chosen. I would also suggest to mention explicitly on p 10 (somewhere around 217-219) that the full range of answers the authors were allowed to share can be found in the OSF data set. 

We have now added more sentences referring to the open dataset to remind and encourage readers to view it. These are now in lines 146, 233, 574 and 722 of the revised manuscript. 

On p 10-11 the authors report that the coders came up with several categories/common themes for the free text analysis. I wasn’t sure how this relates to the different areas of the questions, i.e. whether the referred to “common themes” span all areas or refer to within-area categorization? 

 We have clarified the methods that we used to generate the categories in the methods sections. Specifically, we now stress that “we categorized their free-text responses based on the common themes that they included within each block” (line 236). We then further clarified that the miscellaneous themes were those not systematically extracted, but ones that we interpreted during the coding as repeatedly occurring across blocks, but which were not captured by our within block coding (lines 252-256 in the methods section, and again in lines 474-480 in the results section).

Sometimes there are categories with exemplars in tables (e.g. Table 4) and sometimes only exemplars are reported (e.g. Table 3). In this respect, I found the result presentation a bit inconsistent.

 We have aligned the presentation of all results such that they are now categories with exemplars presented for all tables. To achieve this, we performed another categorisation analysis on the publication questions for Table 5 (which we had originally only performed for the first free-text publication analysis in Table 4), and so have also updated the inter-rater reliability figures (lines 241-243) accordingly). During this additional coding, we realised that interpreting negative results was a larger concern of researchers that originally missed, and so also added this as a final 5th miscellaneous theme to discuss (lines 512-520 in results). 

I think the authors could make a stronger point in discussing the merit or potential use of their results. In line 534-535 they say the results might be used to guide discussions on how animal cognition research might improve in the future, but who should have these discussions - Researchers, universities, journals? For example, could the paper spark fruitful discussion among researchers with opposing views? Does it have the potential to serve as a reference point to inquire with journals, universities, and funding agencies, what their rejection and/or hiring practices are and whether they think their incentives serve the quality of scientific advances, given concerns of the interviewed researchers (e.g. that many researchers report they feel pressured to produce novel & sexy rather than solid scientific outputs)? Does this differ/align with attitudes in other scientific research fields? What initiatives are already out there trying to improve the situation and how can the current paper inform and complement these movements? When I finished reading the manuscript, I was left wondering: “so what?” For me, some more elaboration was missing on how the paper can help improve the situation and who is its target reader group.

 We thank the reviewer for these comments. We have added further discussion about how our results relate to and can impact various initiatives to overcome some of the barriers highlighted in this survey throughout. Specifically, and in line with Reviewer 2’s suggestions, we have added a clearer section on the aims of the survey and potential uses of these data in the introduction (lines 80-91), and at the end of the first paragraph in the discussion (lines 574-587). Throughout the discussion, we have widened our comments from being specific to the results within our survey and have discussed how they relate to topics such as registered reports, pre-prints, funding decisions and the academic incentive structure throughout. 

Typos & formatting issues:

L 161: missing “t” in “…and he last 4 questions…”

L 168: replace “I” with “they”?

L 173: remove “.”

L 258-259: misplaced line break

L 280: misplaced line break

L 347: misplaced line break

Thank-you for identifying these, they have been changed.

Resolution quality of all figures on the pdf was quite bad, sometimes barely readable. .Tiff quality was fine.

We think this is an issue with how the Plos software compiled the manuscript and assume the .Tiff quality will appear in the final version, however we will keep an eye out for this at any further stages. 

Fig 1-5: I am not sure the full questionnaire needs to be in the paper. I would maybe give a formatting example in the paper and provide the full questionnaire in ESM.

For now, we have decided to retain the full survey in the paper, however we are open to putting it in the ESM if the editor prefers. 

Reviewer 2

In this paper, the authors report a timely study on the beliefs of animal cognition researchers as they relate to research practices and bias in their field. The authors surveyed over 200 researchers and find:

- evidence of publication bias,

- that participants feel that results in their field can be biased (due to questionable research practices), although, researchers views of the field in general are more negative than their views of their own practices.

- participants report the importance of replication although many feel that only 60-70% of findings in their field may replicate.

Overall, this paper provides some interesting and important data on how animal cognition researcher view the state of their field in the context of wider scientific issues (and some more field-specific issues, i.e. Morgan's Canon). The paper is well written, with results clearly presented and I think this paper makes an important contribution to our understanding of how animal cognition research operates.

We would like to thank Reviewer 2 for taking the time to provide detailed and constructive comments on the manuscript.

Abstract: Solid abstract covering methods and key results, but missing an overarching conclusion. Putting this evidence together, what do the authors feel the main take-home message is (regarding animal cognition researchers views on these issues?).

We have expanded a couple of sentences at the end of the abstract to better tie together what we think the take-home messages are, and the implications of these data (lines 37-41). 

Introduction: Solid introduction with appropriate referencing throughout. I wonder if a brief paragraph about the study aims (just before the methodology or description of key survey blocks) would help the reader understand exactly what the authors expect to achieve. It is easy to implicitly parse the aims from the intro (i.e., from line 62: "But how effectively these debates are reaching animal cognition researchers in general, and how they are received, has garnered little attention"), but I think a clearer statement would set up the methodology nicely.

We have added a clearer section on the aims of the survey and potential uses of these data in the introduction (lines 80-91).

Methods - Free-text analysis: I am not an expert on qualitative methods, but I think the researchers are describing some form of thematic analysis. The procedure used to develop themes seems fairly robust but I think that results from this qualitative analysis should be reported consistently (see comments below regarding tables in results). Tables (2-7). I am not 100% sure why table 2 is included. It presents a "selection" of some biases that researchers think exist, but how were these examples chosen ? Did they emerge from the most common themes? Are they the clearest and most coherent responses (if so, is this an adequate reason to include them here)? If the point of the table is to merely demonstrate a range of views, that is fine, but a clear description of how these specific views were chosen would be useful. I think Tables 3 and 5 have the same issue, while Tables 4, 6 and 7 are fine (as the method for developing themes has been made clear in the methodology and so the examples make sense). The benefit of reading a summary of themes (e.g. like in table 4) is that readers get an idea of both the range of issues that participants mention of their own accord, as well as the relative frequency of these comments. On the other hand, the quotes summarised in table 2 and 3 are less informative as they are not presented in a systematic fashion that help us pick out themes, and we don't get an idea of how popular these various views are. In short, I feel that the reporting of example text should be consistent and align with the analyses described.

Many thanks for this point. We have now aligned all tables such that they are in the format of the original Tables 4, 6 and 7 (categories, Ns and examples). As we outlined in response to Reviewer 1’s similar comment, this involved performing another categorisation analysis on the publication questions for Table 5 (which had originally only been performed for the first free-text publication analysis in Table 4). Because of this, we have also now updated the inter-rater reliability figures (lines 241-243).

Table 1. Percentage might a more meaningful descriptor for both variables (with Ns in brackets perhaps)? I.e., easier to assess distribution across bins at a glance?

We agree and have added the percentages as an extra row in the table. 

Lines 341- Where the authors report "many researchers also reported" I think an N would be useful (to get a better idea of how this comment generalises to the sample.

We have added the N to this sentence (line 361-362). 

Lines 399-403. When comparing outcomes on ordinal scales wouldn't it be useful to include a non-parametric test to demonstrate a different distribution of results? This isn't necessary for most sections, but it would be useful for a statement like this: "Predominantly, researchers somewhat agreed (34.0%) or somewhat disagreed (30.1%) that their area of animal cognition research would experience a replication crisis if attempts to 401 replicate most of its studies were conducted, however they either somewhat (43.7%) or strongly (29.3%) agreed that some other areas of animal cognition research would experience a replication crisis." Here, the "however" is important as there is a suggestion that the different responses on these measures are meaningful - a Wilcoxon (for example) would tell us whether this difference is likely to be due to chance.

Thank-you for this suggestion. After having given this some thought, we decided against adding a statistical test to complement these statements. To us, there is no clear mechanism by which our respondents would have been answering by chance (i.e., the null that researchers were responding randomly seems implausible). A measure of effect size might be the best way to convey the information, and we feel that the raw numbers and visualisations do this effectively. Nevertheless, to avoid the implication that we have done some statistical test, we have changed the wording from “however they” to “and they”. 

Line 436. Why highlight this quote? Again, I am not against using these examples but the rationale should be made clear.

 For this quote, we added a explanation that we highlighted it because it did not fit one of our themes but (subjectively) interested us (lines 454-455).

Line 454. Why are these themes presented differently from the earlier ones (i.e. in tables with N)? I think that if these themes are important enough to be mentioned they should be presented in a consistent manner (i.e. the number of participants who mentioned these issues should be mentioned, etc.).

 In the methods, and when we present to the results of the miscellaneous themes, we added additional information for how we extracted these (lines 252-256 in the methods, and lines 474-480 in the results). Because these were not collected systematically, as we now highlight in the methods text, we did not collect information on the number of researchers highlighting these. 

Discussion:

Nice to see the authors address general issues with their methods head-on, but I think it would be worthwhile to discuss specific issues in the discussion too and how these may specifically limit interpretations/conclusions. For example, the authors write: "Rather, they must be interpreted considering the likely sampling biases in who participated in our survey and how their answers were limited by the way the questions were asked." Could these sampling biases and their implications be spelled out for us.

We now discuss this further with examples in lines 564-574.

Line 544: Be careful of using qualitative responses as evidence of a "general" beliefs in your participants without providing some qualifiers or justification (e.g. "Importantly, researchers’ qualitative responses suggested that they believe bias not to be uniform across the field, instead reporting that certain topics and researchers may be more likely to be affected by bias than others"). Obviously, the same applies to the use of quantitative results, but this is easier to justify given you are reporting raw proportions.

 Thank-you for picking this up, we are now more careful with our wording throughout, and changed the phrasing of this particular sentence from “Importantly, researchers’ qualitative 545 responses suggested that they believe bias not to be uniform across the field” to “For example, some researchers’ qualitative responses suggested that they believe bias not to be uniform across the field”

Lines 597-599: The authors interpret the qualitative data relating to Morgan's canon using two themes (a) recognised the inherent ambiguity and multiple interpretations of Morgan’s canon, and b) cautioned against a blind application of Morgan’s canon), however, these themes are not used in table 3 (see comment above regarding consistency in qualitative analysis).

 The Morgan’s canon data are now presented in this manner in Table 3, with further explanation in the methods section for how this was generated (lines 248-252): “For the Bias block, we chose to split the results of the open-ended question (“Do you have any other comments about bias in animal cognition research?”) into two tables, as participants’ free-text responses were split between providing examples of biases in animal cognition research and elaborating on their Likert-type responses to the question about Morgan’s canon.”

Discussion - Overall: Good summary of key findings as they relate to the survey questions, and good use of free-response text to qualify (or add nuance) to your interpretation of this data. However, one nice thing about qualitative data is that it can go beyond your questions. I wonder if a section of your discussion could address this - for example, the "miscellaneous themes" extracted in your qualitative analysis only warrant a sentence or two in your discussion, but I feel that these findings help identify the next questions to be put to animal cognition researchers. The conclusion does a good job summarising the main results, but I would like to read about: a) taking the results together what are the take-home messages, b) are researcher practices/beliefs/etc in this field different from other disciplines, c) based upon these results what are the authors recommendations (if any) for the future directions of this type of research, education of researchers to reduce bias, changes to incentives that drive these issues. (in short, I think in the discussion the authors could think bigger)

We have expanded the discussion significantly, and have now included a section for each of the miscellaneous themes. While a discussion of each of these themes could likely make a paper in itself, we have tried to provide an overview with effective references to act as a starting point for the reader. In addition, and in line with Reviewer 1’s comments also, we have provided much further discussion throughout on the relationship between our own findings and initiatives and methods to improve science from across disciplines. Again, we feel such a discussion could make a paper in itself, but have hopefully covered key starting points and references. Finally, we have added in a little more discussion comparing our results with other survey studies of researchers. In addition to the questionable research practices surveys that we cite (refs 42-44) we highlight the study of replication importance in ecology and evolution (ref 41), and the Baker study of concerns about replication across scientific disciplines (ref 61).

Very minor issues, typos, etc.:

Line 74: "In the current study, we used surveyed researchers’ attitudes"

Lines 153-158. The presentation order here is different from the order in the intro- probably best to be consistent unless there is a good reason.

Line 161. "he" should be "the"

Line 527-529: The following sentence is a bit long and could benefit from being rephrased: "Specifically, we do not believe that these data are a very accurate or representative data of all animal cognition researchers’ beliefs, or very accurate estimates of, for example, the absolute rate of questionable research practice use in the field (see e.g. 42])."

Line 573/574/577. Should "Figs" be "figures"?

 We thank the reviewer for highlighting these, all of which have been changed.

---

## [Editor Report · Decision Letter 1]

11 Aug 2021

The Hidden Side of Animal Cognition Research: Scientists’ Attitudes Toward Bias, Replicability and Scientific Practice

PONE-D-21-11911R1

Dear Dr. Farrar,

Thank you for taking the time to revise your manuscript. I am satisfied with your responses to the reviewer feedback and the associated changes in the manuscript, and am therefore pleased to inform you that your manuscript has been judged scientifically suitable for publication and will be formally accepted for publication once it meets all outstanding technical requirements.

Kind regards,

Mark Nielsen, Ph.D.

Academic Editor

PLOS ONE

---

## [Editor Report · Acceptance letter]

12 Aug 2021

PONE-D-21-11911R1 

The Hidden Side of Animal Cognition Research: Scientists’ Attitudes Toward Bias, Replicability and Scientific Practice 

Dear Dr. Farrar:

I'm pleased to inform you that your manuscript has been deemed suitable for publication in PLOS ONE. Congratulations! Your manuscript is now with our production department. 

Kind regards, 

on behalf of

Dr. Mark Nielsen 

Academic Editor

PLOS ONE